# Nuclear SUN1 stabilizes endothelial cell junctions via microtubules to regulate blood vessel formation

Danielle B Buglak[1†], Pauline Bougaran[2], Molly R Kulikauskas[1], Ziqing Liu[2], Elizabeth Monaghan-Benson[3], Ariel L Gold[2], Allison P Marvin[2], Andrew Burciu[2], Natalie T Tanke[1], Morgan Oatley[2], Shea N Ricketts[4], Karina Kinghorn[1], Bryan N Johnson[2], Celia E Shiau[2], Stephen Rogers[2], Christophe Guilluy[3], Victoria L Bautch[1,2,5]*

[1]Curriculum in Cell Biology and Physiology, The University of North Carolina at Chapel Hill, Chapel Hill, United States; [2]Department of Biology, The University of North Carolina at Chapel Hill, Chapel Hill, United States; [3]Department of Molecular Biomedical Sciences, College of Veterinary Medicine, North Carolina State University, Raleigh, United States; [4]Department of Pathology, The University of North Carolina at Chapel Hill, Chapel Hill, United States; [5]McAllister Heart Institute, The University of North Carolina at Chapel Hill, Chapel Hill, United States

*For correspondence:
bautch@med.unc.edu

Present address: †National Heart, Lung, and Blood Institute, National Institutes of Health, Bethesda, United States

**Abstract** Endothelial cells line all blood vessels, where they coordinate blood vessel formation and the blood-tissue barrier via regulation of cell-cell junctions. The nucleus also regulates endothelial cell behaviors, but it is unclear how the nucleus contributes to endothelial cell activities at the cell periphery. Here, we show that the nuclear-localized linker of the nucleoskeleton and cytoskeleton (LINC) complex protein SUN1 regulates vascular sprouting and endothelial cell-cell junction morphology and function. Loss of murine endothelial *Sun1* impaired blood vessel formation and destabilized junctions, angiogenic sprouts formed but retracted in SUN1-depleted sprouts, and zebrafish vessels lacking Sun1b had aberrant junctions and defective cell-cell connections. At the cellular level, SUN1 stabilized endothelial cell-cell junctions, promoted junction function, and regulated contractility. Mechanistically, SUN1 depletion altered cell behaviors via the cytoskeleton without changing transcriptional profiles. Reduced peripheral microtubule density, fewer junction contacts, and increased catastrophes accompanied SUN1 loss, and microtubule depolymerization phenocopied effects on junctions. Depletion of GEF-H1, a microtubule-regulated Rho activator, or the LINC complex protein nesprin-1 rescued defective junctions of SUN1-depleted endothelial cells. Thus, endothelial SUN1 regulates peripheral cell-cell junctions from the nucleus via LINC complex-based microtubule interactions that affect peripheral microtubule dynamics and Rho-regulated contractility, and this long-range regulation is important for proper blood vessel sprouting and junction integrity.

## Editor's evaluation

Endothelial cells lining the inner side of blood vessels constitute the blood-tissue barrier via regulation of cell-cell junctions. The cell nucleus regulates endothelial cell behaviors, but it is unclear how the nucleus contributes to endothelial cell activities at the cell membrane. This study for the first time demonstrates that nuclear membrane protein SUN1 stabilizes endothelial cell-cell junctions far from the nucleus via regulation of microtubule dynamics and Rho GEF-H1 signaling, revealing long-range cellular communication important for vascular development and endothelial barrier function.

## Introduction

Blood vessels form and expand via sprouting angiogenesis, a dynamic process whereby endothelial cells migrate from pre-existing vessels to form new conduits (*Carmeliet and Jain, 2011*; *Wacker and Gerhardt, 2011*; *Kushner and Bautch, 2013*; *Bautch and Caron, 2015*). During angiogenesis, endothelial cell-cell junctions destabilize and rearrange to allow for repolarization and migration toward pro-angiogenic cues (*Esser et al., 1998*; *Dejana, 2004*; *Blum et al., 2008*). Specifically, the endothelial cell adherens junction protein VE-cadherin is required for vascular sprouting and viability (*Carmeliet et al., 1999*; *Montero-Balaguer et al., 2009*; *Sauteur et al., 2014*; *Szymborska and Gerhardt, 2018*). As vessels mature, endothelial cell junctions stabilize and form a functional barrier that regulates egress of fluid and oxygen; barrier dysfunction leads to increased permeability and severe disease (*Claesson-Welsh, 2015*; *Rho et al., 2017*; *Claesson-Welsh et al., 2021*). Thus, regulation of endothelial cell junction stability is important developmentally and for vascular homeostasis.

Adherens junctions are key to integrating and regulating both external and internal cellular inputs from multiple sources, including the microtubule and actin cytoskeletons (*Ligon et al., 2001*; *Shaw et al., 2007*; *Bellett et al., 2009*; *Dejana and Vestweber, 2013*; *Abu Taha and Schnittler, 2014*). For example, increased actomyosin contractility destabilizes endothelial cell adherens junctions, and disorganized junctional actin accompanies VE-cadherin loss (*Huveneers et al., 2012*; *Sauteur et al., 2014*; *Angulo-Urarte et al., 2018*). VE-cadherin loss also changes microtubule dynamics, and disruption of microtubule dynamics destabilizes junctions and barrier function (*Komarova et al., 2012*). Coordination of inputs from the actin and microtubule cytoskeletons regulates endothelial cell barrier integrity and sprouting dynamics via small GTPases (*Birukova et al., 2006*; *Mavria et al., 2006*; *Sehrawat et al., 2008*; *Sehrawat et al., 2011*; *Wimmer et al., 2012*; *Szymborska and Gerhardt, 2018*). In particular, RhoA signaling is regulated by microtubules via GEF-H1, a RhoGEF that is inactive while bound to microtubules and activated upon release, leading to RhoA activation, increased actomyosin contractility, and changes to endothelial cell barrier function (*Krendel et al., 2002*; *Birukova et al., 2006*; *Birkenfeld et al., 2008*). However, how the endothelial cell nucleus affects these processes is poorly understood.

The nucleus is usually found far from the cell periphery and junctions, yet it is important for functions critical to angiogenesis and vascular remodeling, such as polarity, migration, and mechanotransduction (*Tkachenko et al., 2013*; *Guilluy et al., 2014*; *Graham et al., 2018*), and perturbations of some nuclear membrane proteins affect transcriptional profiles (*Li et al., 2017*; *May and Carroll, 2018*; *Carley et al., 2021*). The linker of the nucleoskeleton and cytoskeleton (LINC) complex is comprised of both SUN (Sad1p, UNC-84) and KASH (Klarsicht, ANC-1, Syne/Nesprin Homology) proteins (*Starr and Fridolfsson, 2010*) that function as a bridge between the nucleus and the cytoskeleton, and also link through subnuclear lamin filaments to chromatin (*Haque et al., 2006*). SUN proteins localize to the inner nuclear membrane and bind KASH proteins, or nesprins, from their C-terminus and lamins at their N-terminus, thus providing a structural link from the nuclear cortex to the cellular cytoskeleton (*Padmakumar et al., 2005*; *Haque et al., 2006*; *McGee et al., 2006*; *Stewart-Hutchinson et al., 2008*). Nesprins are long spectrin-rich proteins localized to the outer nuclear envelope that bind SUN proteins via their C-terminus while N-terminally interacting indirectly with microtubules (via various motor proteins such as dynein and kinesin) and intermediate filaments (via plectins), and directly with actin via calponin homology domains (*Ketema et al., 2007*; *Meyerzon et al., 2009*; *Zhang et al., 2009*; *Fridolfsson et al., 2010*; *Starr and Fridolfsson, 2010*). Two mammalian SUN proteins are ubiquitously expressed, and based on functional consequences of SUN manipulations it has been posited that SUN1 regulates microtubule-based functions while SUN2 coordinates actin regulation (*Zhu et al., 2017*). However, in vitro binding studies do not reveal a SUN-nesprin specificity to account for this bias (*Stewart-Hutchinson et al., 2008*; *Ostlund et al., 2009*), so how complexes are assembled and sorted in cells is unclear. It is well established that the LINC complex integrates external inputs sensed by focal adhesions, such as substrate stiffness, to regulate transcription (*Carley et al., 2022*), but how the LINC complex relays signals from the nucleus to the cell periphery is less understood.

The LINC complex functions in cultured endothelial cells, as knockdown (KD) of nesprin-3 leads to impaired endothelial polarity under flow (*Morgan et al., 2011*), while nesprin-2 and lamin A regulate proliferation and apoptosis in endothelial cells exposed to shear stress (*Han et al., 2015*). Depletion of nesprin-1 alters nuclear tension (*Chancellor et al., 2010*; *Anno et al., 2012*) and force application to isolated nuclei via nesprin-1 alters stiffness (*Guilluy et al., 2014*), while KD of nesprin-1 or nesprin-2

leads to reduced collective endothelial cell migration (*Chancellor et al., 2010*; *King et al., 2014*). Recent work showed compromised matrix adhesion and barrier function of cultured endothelial cells using a dominant negative KASH (*Denis et al., 2021*). However, less is understood about the roles of the SUN proteins in endothelial cell function or how the LINC complex functions in vessels in vivo (*Salvador and Iruela-Arispe, 2022*).

The LINC complex is required for viability in vivo. Loss of both mammalian *Sun* genes is embryonic lethal due to impaired neuronal nuclear migration required for proper neuronal differentiation (*Lei et al., 2009*; *Zhang et al., 2009*). Global *Sun2* loss affects epidermal nuclear positioning and cell adhesion, leading to alopecia (*Stewart et al., 2015*), while global loss of both *Sun* genes impairs epidermal differentiation due to altered integrin signaling (*Carley et al., 2021*). A role for the LINC complex in mechanotransduction in vivo is suggested by findings that perturbations in mechanically active skeletal and cardiac muscle affect function (*Zhang et al., 2005*; *Zhang et al., 2010*; *Lei et al., 2009*; *Banerjee et al., 2014*; *Stroud et al., 2017*; *Zhou et al., 2017*; *van Ingen and Kirby, 2021*). However, while global deletion of multiple LINC components that disrupt the entire complex highlight its importance (*Lei et al., 2009*; *Zhang et al., 2009*; *Carley et al., 2021*), these studies do not reveal functions of individual LINC components in specific tissues. Whether the SUN proteins cell autonomously regulate the vascular endothelium, which is also mechanically active due to outward pressure and shear stress from blood flow, has not been explored.

Mutations in *LMNA* (lamin A/C) cause a premature aging syndrome linked to cardiovascular defects (*Capell and Collins, 2006*). The LINC complex protein SUN1 overaccumulates in the nucleus in this disease (*Chen et al., 2012*), and cellular defects due to the *LMNA* mutation are rescued by reduced levels of SUN1 protein, highlighting a potential function for SUN1 in the disease pathology (*Chen et al., 2012*; *Chang et al., 2019*). Here, we present an in-depth analysis of how the LINC complex component SUN1 affects blood vessel development and function in endothelial cells in vivo. We found that *Sun1* cell autonomously regulates blood vessel sprouting and junction properties in vivo, consistent with SUN1 regulating endothelial cell functions via adherens junction activity. In primary endothelial cells, nuclear SUN1 coordinates peripheral microtubule dynamics that in turn regulate peripheral RhoGEF activation and junction stability. Thus, nuclear SUN1 that resides far from cell junctions regulates cell-cell communication and blood vessel sprouting via a novel microtubule-based integration pathway from the nucleus to the cell periphery.

## Results

### The nuclear LINC protein SUN1 regulates vascular development

The LINC complex is important for cell migration (*Chancellor et al., 2010*; *King et al., 2014*; *Denis et al., 2021*), and blood vessel formation involves extensive endothelial cell migration; thus, we hypothesized that the LINC complex regulates angiogenic sprouting. Because mutations in endothelial cell *LMNA* causative for human cardiovascular disease are associated with expression changes in the LINC protein SUN1 (*Chen et al., 2012*), and because *Sun1* has not been functionally analyzed in the vascular endothelium in vivo, we first asked whether SUN1 is required for vascular development. Utilizing a mouse line carrying a conditional *Sun1* allele that we generated from *Sun1^{tm1a}* 'knockout first' mice (*Figure 1—figure supplement 1A–B*; *Skarnes et al., 2011*), *Sun1^{fl/fl}* mice were bred to *Sun1^{fl/+};Cdh5-CreERT2/+* mice to generate *Sun1*iECKO (inducible endothelial cell knockout) mice with both endothelial cell-selective and temporal control over *Sun1* excision. Examination of lung DNA, which is rich in endothelial cells, revealed appropriate excision in vivo (*Figure 1—figure supplement 1C*).

The retinal vasculature of *Sun1*iECKO pups injected with tamoxifen at P (postnatal day) 1–3 and harvested at P7 had significantly reduced radial expansion relative to littermate controls (*Figure 1A–C*, *Figure 1—figure supplement 1D*), consistent with a role for *Sun1* in vascular development. *Sun1*iECKO retinas also had increased density at the vascular front (*Figure 1—figure supplement 1E*), consistent with defects in sprouting that prevent expansion and thus increase density (*Hellström et al., 2007*; *Ricard et al., 2012*; *Angulo-Urarte et al., 2018*). Since vessel densities in the plexus ahead of arteries and veins exhibit heterogeneity, we measured by area and found increased density in the plexus ahead of both arteries and veins in *Sun1*iECKO retinas (*Figure 1B and D*). We did not find differences in nuclear shape between *Sun1*iECKO and control retinas, though we did observe

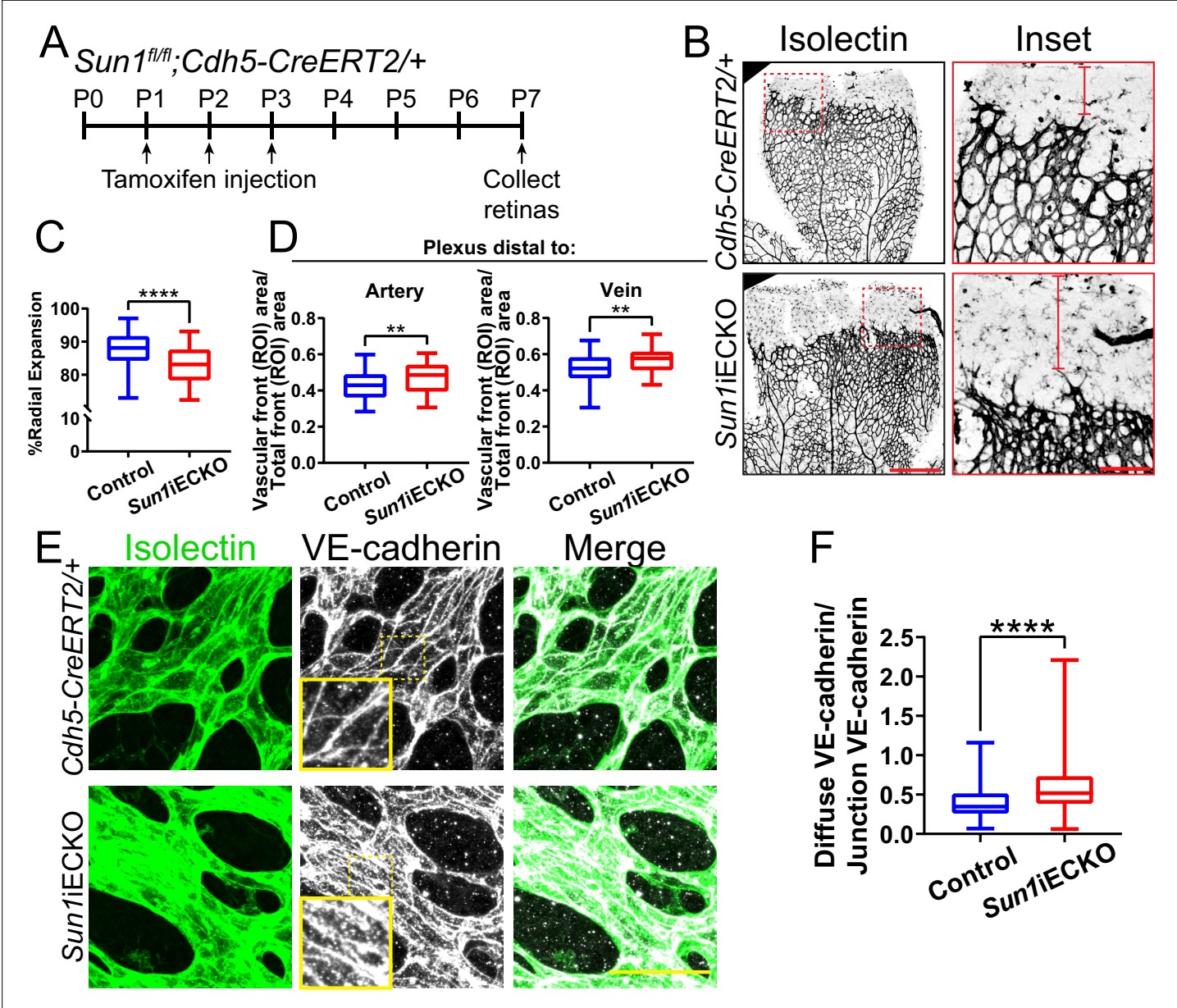

**Figure 1.** The nuclear LINC protein SUN1 regulates vascular development. (**A**) Schematic of tamoxifen-induced excision of exon 4 of *Sun1* in pups from cross of *Sun1^{fl/fl}* × *Sun1^{fl/+}*;*Cdh5-CreERT2* mice. (**B**) Representative images of postnatal day (P)7 mouse retinas of indicated genotypes, stained for IB4 (isolectin). Scale bar, 500 μm. Inset shows vascular plexus ahead of vein. Red line shows expansion of vascular front. Scale bar inset, 150 μm. (**C**) Quantification of vessel network radial expansion in (**B**). n=186 ROIs from 44 retinas (controls) and 63 ROIs from 16 retinas (*Sun1*iECKO) from six independent litters. ****, p<0.0001 by Student's two-tailed unpaired *t*-test. (**D**) Quantification of vascular density ahead of either arteries or veins. n=87 ROIs (controls, artery), 38 ROIs (*Sun1*iECKO, artery), 84 ROIs (controls, vein), and 37 ROIs (*Sun1*iECKO, vein) from 27 retinas (controls) and 12 retinas (*Sun1*iECKO) from three independent litters. **, p<0.01 by Student's two-tailed unpaired *t*-test. (**E**) Representative images of IB4 (isolectin) (green, vessels) and VE-cadherin (white, junctions) staining in P7 retinas of indicated genotypes. Scale bar, 50 μm. (**F**) Quantification of disorganized VE-cadherin as shown in (**E**). n=160 junctions (10 retinas, controls) and 160 junctions (10 retinas, *Sun1*iECKO). ****, p<0.0001 by Student's two-tailed unpaired *t*-test. For all graphs, boxes represent the upper quartile, lower quartile, and median; whiskers represent the minimum and maximum values.

The online version of this article includes the following source data and figure supplement(s) for figure 1:

**Figure supplement 1.** Loss of *Sun1* in the postnatal retina leads to altered sprouting and junction integrity.

**Figure supplement 1—source data 1.** Agarose ethidium bromide gel showing PCR bands specific for WT or *Sun1^{fl}* allele (left) or PCR band for the presence of the *Cdh5-CreERT2* allele (right).

**Figure supplement 1—source data 2.** Agarose ethidium bromide gel showing PCR band specific for the excised *Sun1* allele.

some nuclear crowding in *Sun1*iECKO retinas (*Figure 1—figure supplement 1F*), consistent with the increase in vascular density. However, there was no significant difference in nuclear area relative to vessel area between *Sun1*iECKO and control retinas (*Figure 1—figure supplement 1F–G*). Because adherens junction dynamics regulate vascular sprouting (*Sauteur et al., 2014*), this mutant phenotype suggested that endothelial cell-cell junctions were affected by loss of *Sun1*. VE-cadherin localization, a readout of junction integrity (*Huveneers et al., 2012*; *Bentley et al., 2014*; *Wylie et al., 2018*; *Vion et al., 2020*), was significantly less linear and more punctate in *Sun1*iECKO vessels, indicating increased adherens junction turnover and junction instability (*Figure 1E–F*). Dextran injection was used to functionally evaluate the effects of *Sun1* loss on endothelial cell junction integrity in vivo, and *Sun1*iECKO mice had increased signal in the surrounding tissue compared to controls (*Figure 1—figure supplement 1H*), suggesting increased vascular leakage in the postnatal retina. Together, these data indicate a specific role for SUN1 in angiogenic sprouting and endothelial cell-cell junctions in vivo.

## Nuclear SUN1 is required for sprouting angiogenesis

*Sun1* loss disrupts vascular development in the postnatal mouse retina (*Figure 1*), but this tissue is not amenable to long-term live image analysis. To query dynamic aspects of angiogenic sprouting, which occurs via regulated changes in endothelial adherens junction stability (*Sauteur et al., 2014*; *Angulo-Urarte et al., 2018*; *Wylie et al., 2018*), we utilized a 3D sprouting model (*Nakatsu and Hughes, 2008*) coupled with temporal image acquisition. Reduced levels of endothelial cell SUN1 via siRNA KD (*Figure 2—figure supplement 1A–B*) led to significantly decreased sprout length and branching (*Figure 2A–C*), reminiscent of the decreased radial expansion of *Sun1*iECKO retinal vessels described above. SUN1 depletion did not significantly influence the proportion of EdU-labeled or Ki67 stained cells (*Figure 2—figure supplement 1C–F*), indicating that the abnormal sprouting and branching is not downstream of reduced proliferation. Live-cell imaging revealed that control sprouts typically elongated over time, with little retraction once they extended from the bead (*Figure 2D–E*, *Video 1*). In contrast, SUN1 KD sprouts retracted more often, and many mutant sprouts collapsed partially or completely (*Figure 2D–E*, *Video 2*). SUN1 KD sprouts also showed a more diffuse VE-cadherin junction pattern (*Figure 2F–G*), similar to those of *Sun1*iECKO mice and indicative of disorganized junctions. Thus, SUN1 is required for proper vascular sprout dynamics and morphology, and reduced sprout length and branching are likely downstream of excess sprout retractions and perturbed junctions in SUN1-depleted vessels.

The LINC complex is important for mechanotransduction in muscle fibers with high mechanical loads (*van Ingen and Kirby, 2021*), and sprouting angiogenesis is regulated by mechanical forces arising from blood pressure and blood flow (*Huang et al., 2003*). To determine whether SUN1 also regulates sprouting dynamics under laminar flow in vivo, we analyzed embryonic zebrafish using a *Tg(fli:LifeAct-GFP)* reporter that labels the endothelial actin cytoskeleton. Zebrafish have two *sun1* genes, *sun1a* and *sun1b*; the SUN domain of Sun1b is more homologous to human SUN1, and *sun1b* is more highly expressed in cardiovascular tissue (*Bastian et al., 2021*), so this gene was chosen for manipulation. Sun1b depletion in zebrafish embryos via morpholino (MO) injection led to significantly increased numbers of shorter endothelial cell filopodia at 33–34 hpf (hours post fertilization) in the inter-segmental vessels (ISVs) that sprout from the dorsal aorta and connect to the dorsal longitudinal anastomotic vessel (DLAV) (*Figure 3A–C*). Like the morphants, fish carrying a point mutation in the *sun1b* gene leading to a premature stop codon (*sun1b*[sa33109], see Materials and Methods for details) had shorter filopodia, although filopodia numbers were unchanged in the mutant background (*Figure 3D–F*). Because increased filopodia are typically seen in actively migrating and sprouting endothelial cells (*DeLisser, 2011*), these changes are consistent with Sun1b regulating endothelial cell activation in developing zebrafish vessels exposed to physiological flow forces.

Next, *sun1b* morphant fish were imaged from 26 to 36 hpf to determine the effects of Sun1b depletion on vascular sprouting in vivo. In controls, the ISVs sprouted toward the dorsal plane and connected to the DLAV between 32 and 36 hpf (*Figure 3G–H*, *Video 3*). In contrast, numerous ISVs either failed to reach the DLAV or made aberrant connections in *sun1b* morphant fish (*Figure 3G–H*, *Video 4*). Staining for the tight junction protein ZO-1 revealed less linear and more abnormally shaped junctions in *sun1b* morphant fish, independent of ISV-DLAV anastomosis (*Figure 3I–J*). These results complement the 3D sprouting analysis and show that the nuclear LINC complex protein SUN1 is

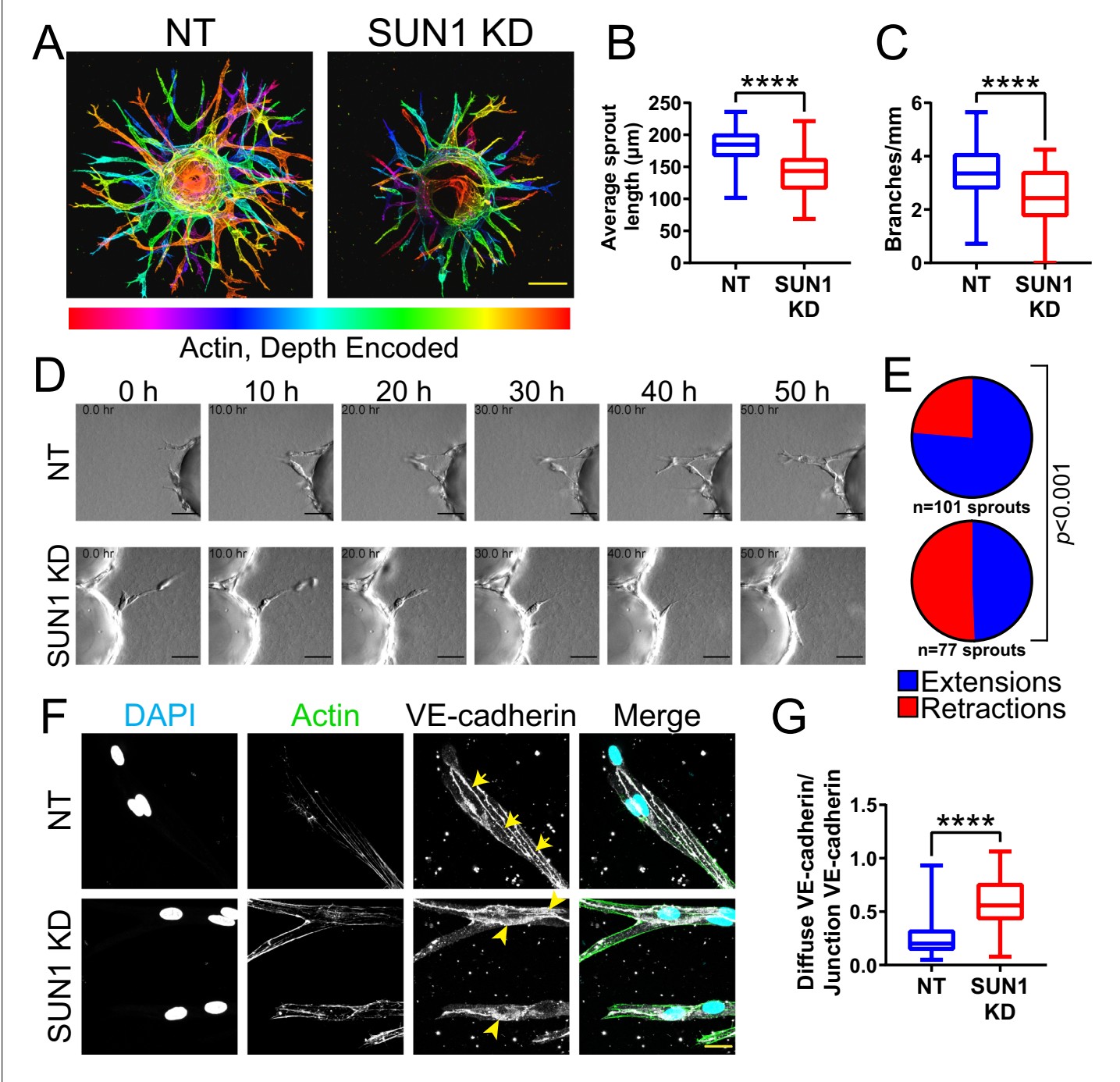

**Figure 2.** Nuclear SUN1 is required for sprouting angiogenesis. (**A**) Representative images of human umbilical vein endothelial cells (HUVEC) with indicated siRNAs in 3D angiogenic sprouting assay. Sprouts were stained for Phalloidin (actin) and then depth encoded such that cooler colors are further in the Z-plane and warmer colors are closer in the Z-plane. Scale bar, 100 µm. (**B**) Quantification of average sprout length of 3D angiogenic sprouts shown in (**A**). n=42 beads (non-targeting [NT]) and 43 beads (SUN1 knockdown [KD]) compiled from five replicates. ****, p<0.0001 by Student's two-tailed unpaired *t*-test. (**C**) Quantification of branches/mm of 3D angiogenic sprouts shown in (**A**). n=41 beads (NT) and 43 beads (SUN1 KD) compiled from five replicates. ****, p<0.0001 by Student's two-tailed unpaired *t*-test. (**D**) Stills from *Video 1* and *Video 2* showing sprouting dynamics of HUVEC with indicated siRNAs over 50 hr. Scale bar, 50 µm. (**E**) Quantification of HUVEC sprout extensions and retractions shown in (**D**). n=101 sprouts (NT) and 77 sprouts (SUN1 KD) compiled from three replicates. p<0.001 by $\chi^2$ analysis. (**F**) Representative images of HUVEC with indicated siRNAs and stained with indicated antibodies in the 3D sprouting angiogenesis assay. Endothelial cells were stained for DAPI (cyan, DNA), Phalloidin (green, actin), and VE-cadherin (white, junctions). Arrows indicate normal junctions; arrowheads indicate abnormal junctions. Scale bar, 20 µm. (**G**) Quantification of disorganized VE-cadherin as shown in (**F**). n=32 junctions (NT) and 30 junctions (SUN1 KD) compiled from two replicates. ****, p<0.0001 by Student's

*Figure 2 continued on next page*

*Figure 2 continued*

two-tailed unpaired *t*-test. For all graphs, boxes represent the upper quartile, lower quartile, and median; whiskers represent the minimum and maximum values.

The online version of this article includes the following figure supplement(s) for figure 2:

**Figure supplement 1.** SUN1 is nuclear localized in endothelial cells and does not regulate proliferation.

important in regulating endothelial cell sprouting dynamics and junction morphology under flow forces in vivo.

## SUN1 stabilizes endothelial cell-cell junctions and regulates junction integrity

SUN1 loss or depletion in mouse, zebrafish, and 3D sprouting models resulted in abnormal endothelial cell-cell junctions and sprouting behaviors. Thus, we examined more rigorously the hypothesis that SUN1 regulates endothelial cell junction stability and morphology. Primary human umbilical vein endothelial cells (HUVEC) in confluent monolayers had more serrated cell-cell junctions without altered levels of VE-cadherin protein expression after SUN1 depletion, indicative of destabilized junctions (*Figure 4A*, *Figure 4—figure supplement 1A–B*). Destabilized endothelial cell adherens junctions are associated with impaired junction integrity, so we measured electrical resistance across endothelial monolayers using real time cell analysis (RTCA) that provides an impedance value positively correlated with junction integrity. SUN1 KD endothelial cells had reduced electrical resistance compared to controls (*Figure 4B–C*), indicative of functional consequences to endothelial junctions and consistent with the increased dextran leakage in vivo. Endothelial cells in vivo are exposed to blood flow that remodels endothelial junctions (*Seebach et al., 2000*; *Yang et al., 2020*). SUN1-depleted HUVEC exposed to laminar shear stress for 72 hr elongated and aligned properly, but adherens junctions were more serrated and allowed for significantly more matrix exposure as assessed by a biotin labeling assay (*Dubrovskyi et al., 2013*) under both static and flow conditions (*Figure 4D–F*, *Figure 4—figure supplement 1C–D*). These data are consistent with the finding that a KASH dominant negative construct disrupts endothelial electrical impedance (*Denis et al., 2021*) and support the hypothesis that nuclear SUN1 regulates junction integrity via effects on endothelial cell junction stability and morphology.

Dysfunctional cell-cell junctions can result from abnormal junction formation or the inability of formed junctions to stabilize. To determine how

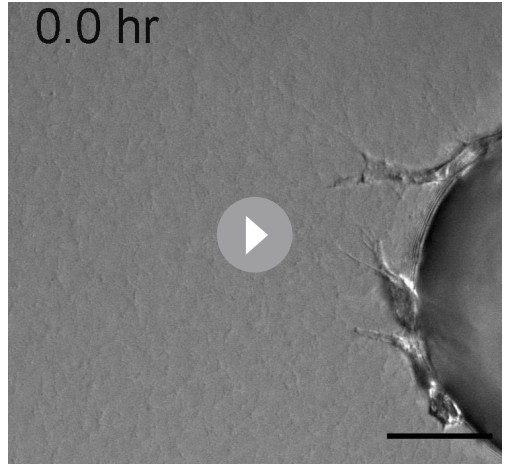

**Video 1.** Control endothelial cells elongate in 3D sprouting assay. 3D sprouting angiogenesis of control (non-targeting [NT]) human umbilical vein endothelial cells (HUVEC) over 60 hr, showing elongation of NT sprouts. Scale bar, 50 μm. Frames acquired every 30 min.

https://elifesciences.org/articles/83652/figures#video1

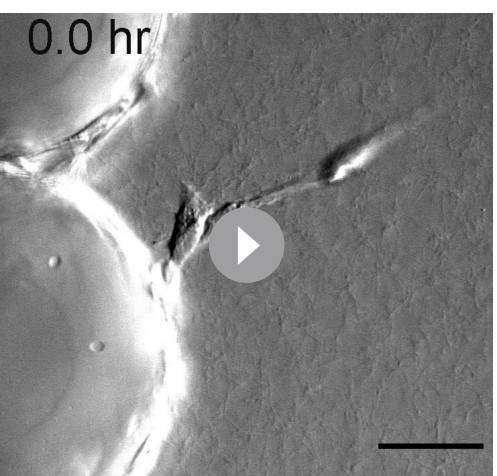

**Video 2.** SUN1-depleted endothelial cells retract in 3D sprouting assay. 3D sprouting angiogenesis of SUN1 knockdown (KD) sprouts over 60 hr, showing retraction of SUN1 KD sprouts. Scale bar, 50 μm. Frames acquired every 30 min.

https://elifesciences.org/articles/83652/figures#video2

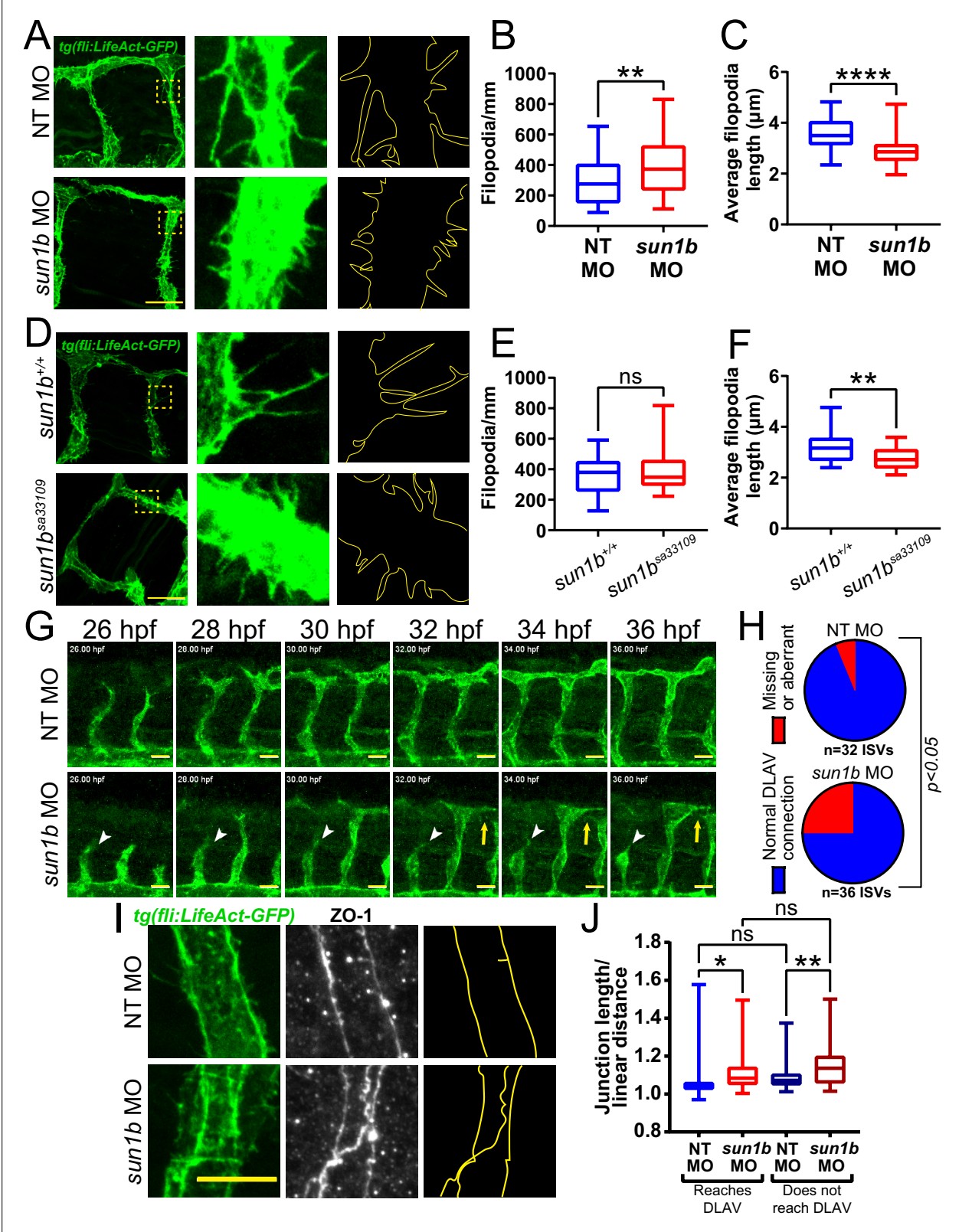

**Figure 3.** SUN1 regulates actin dynamics and angiogenic sprout extension in vivo. (**A**) Representative images of zebrafish embryos at 34 hpf (hours post fertilization) with indicated morpholino treatments; anterior to left. *Tg(fli:LifeAct-GFP)* (green, vessels). Insets show inter-segmental vessels (ISVs) with filopodia, outlines highlighted to show filopodia. Scale bar, 20 μm. (**B**) Quantification of filopodia number shown in (**A**). n=39 ROIs (15 fish, non-targeting [NT] morpholino [MO]) and 56 ROIs (20 fish, *sun1b* MO) compiled from three replicates. **, p<0.01 by Student's two-tailed unpaired *t*-test. (**C**)

*Figure 3 continued on next page*

*Figure 3 continued*

Quantification of average filopodia length shown in (**A**). n=39 ROIs (15 fish, NT MO) and 56 ROIs (20 fish, *sun1b* MO) compiled from three replicates. ****, p<0.0001 by Student's two-tailed unpaired *t*-test. (**D**) Representative images of zebrafish embryos at 34 hpf with indicated genotypes; anterior to left. *Tg(fli:LifeAct-GFP)* (green, vessels). Insets show ISVs with filopodia, outlines highlighted to show filopodia. Scale bar, 20 μm. (**E**) Quantification of filopodia number shown in (**D**). n=27 ROIs (9 fish, *sun1*$^{+/+}$) and 30 ROIs (10 fish, *sun1*$^{sa33109}$) compiled from two replicates. ns, not significant by Student's two-tailed unpaired *t*-test. (**F**) Quantification of average filopodia length shown in (**D**). n=27 ROIs (9 fish, *sun1*$^{+/+}$) and 30 ROIs (10 fish, *sun1*$^{sa33109}$) compiled from two replicates. **, p<0.01 by Student's two-tailed unpaired *t*-test. (**G**) Stills from *Video 3* and *Video 4* showing ISV sprouting from 26 to 36 hpf in zebrafish embryos with indicated morpholino treatment; anterior to left. *Tg(fli:LifeAct-GFP)* (green, vessels). White arrowhead points to ISV that does not extend or connect to dorsal longitudinal anastomotic vessel (DLAV). Yellow arrow points to ISV that extends but does not connect to DLAV. Scale bar, 20 μm. (**H**) Quantification of ISV connection to DLAV shown in (**G**). n=32 ISVs (6 fish, NT MO) and 36 ISVs (6 fish, *sun1b* MO) compiled from two replicates. p<0.05 by $\chi^2$ analysis. (**I**) Representative images of zebrafish embryos at 34 hpf with indicated morpholino treatments; anterior to left. *Tg(fli:LifeAct-GFP)* (green, vessels); ZO-1 (white junctions). Outlines highlighted to show junction shapes. Scale bar, 10 μm. (**J**) Quantification of junction morphology shown in (**I**). n=80 junctions that reach DLAV and 68 junctions that do not reach DLAV (21 fish, NT MO) and 47 junctions that reach DLAV and 37 junctions that do not reach DLAV junctions (22 fish, *sun1b* MO). ns, not significant; *, p<0.05; **, p<0.01 by two-way ANOVA with Tukey's multiple comparisons test. For all graphs, boxes represent the upper quartile, lower quartile, and median; whiskers represent the minimum and maximum values.

SUN1 functionally regulates endothelial junctions, adherens junctions were disassembled via Ca$^{2+}$ chelation, then reformed upon chelator removal. Junctions were measured using line scans of VE-cadherin intensity along the cell-cell junctions (*Figure 4—figure supplement 1E*), such that junctions with a linear VE-cadherin signal (stable) had a higher value than those with more serrated patterns (destabilized). No significant difference between SUN1 KD and control junctions was seen at early times post-washout, indicating that SUN1 depletion does not affect adherens junction formation (*Figure 4G–H*). However, later times post-washout revealed a significant increase in serrated junctions and gaps between endothelial cells in SUN1 KD monolayers relative to controls (*Figure 4G–H*). Consistent with these findings, SUN1 KD endothelial cells also had increased VE-cadherin internalization at steady state, consistent with actively remodeling

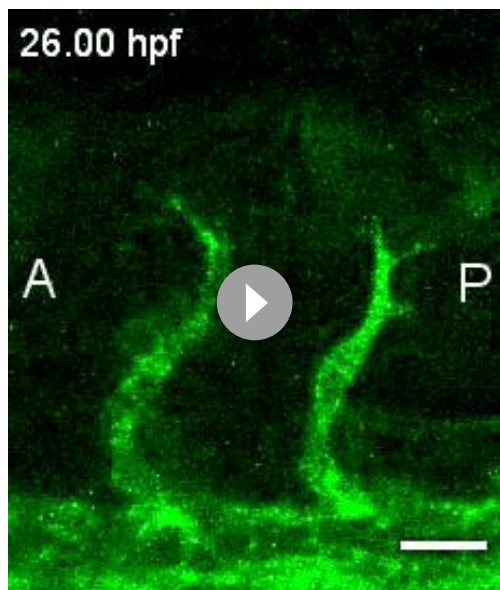

**Video 3.** Control zebrafish have normal inter-segmental vessel (ISV) growth. Movie taken from 26 to 36 hpf (hours post fertilization) in *Tg(fli:LifeAct-GFP)* zebrafish embryos injected with a non-targeting (NT) morpholino, showing elongation of ISVs and connection to the dorsal longitudinal anastomotic vessel (DLAV). A, anterior; P, posterior. Scale bar, 20 μm. Frames acquired every 15 min.

https://elifesciences.org/articles/83652/figures#video3

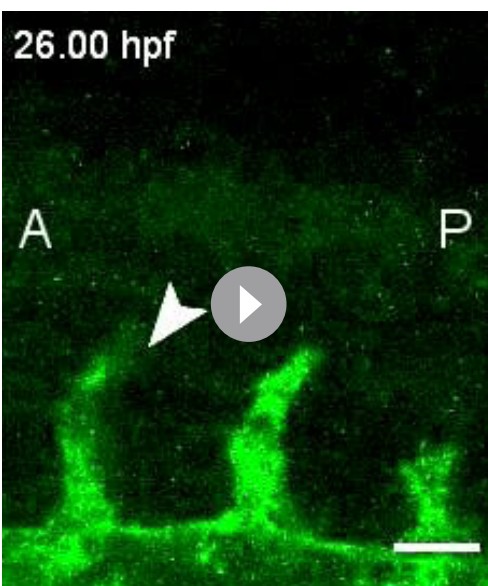

**Video 4.** Loss of SUN1 in zebrafish leads to abnormal inter-segmental vessel (ISV) growth. Movie taken from 26 to 36 hpf in *Tg(fli:LifeAct-GFP)* zebrafish embryos injected with a *sun1b* morpholino, showing an ISV that fails to elongate and connect to the dorsal longitudinal anastomotic vessel (DLAV) and an ISV that elongates but does not connect to the DLAV. A, anterior; P, posterior. Arrow points to ISV that does not elongate. Scale bar, 20 μm. Frames acquired every 15 min.

https://elifesciences.org/articles/83652/figures#video4

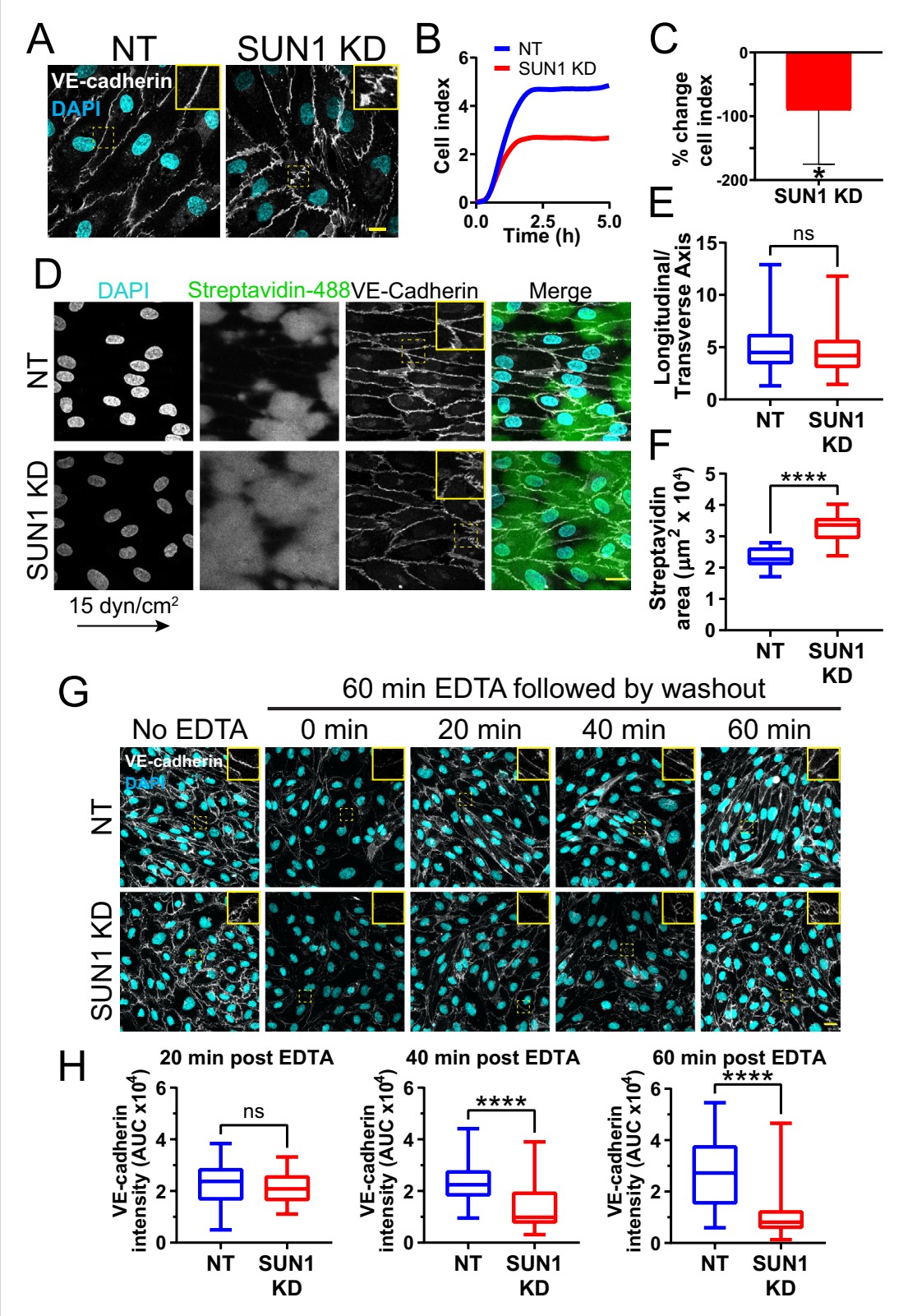

**Figure 4.** SUN1 stabilizes endothelial cell-cell junctions and regulates junction integrity. (**A**) Representative images of human umbilical vein endothelial cells (HUVEC) with indicated knockdowns (KD) in monolayers. Endothelial cells were stained for DAPI (cyan, DNA) and VE-cadherin (white, junctions). Insets show junctions. Scale bar, 10 µm. (**B**) Representative graph of impedance measured by real time cell analysis (RTCA). (**C**) Quantification of % change in cell index for RTCA measured at 5 hr. Normalized to non-targeting (NT) cell index. n=5 replicates. *, p<0.05 by Student's two-tailed unpaired

*Figure 4 continued on next page*

*Figure 4 continued*

t-test. (**D**) Representative images of HUVEC with indicated siRNAs plated on biotinylated fibronectin and exposed to 15 dyn/cm² shear stress for 72 hr then treated with streptavidin. Endothelial cells were stained for DAPI (cyan, DNA), streptavidin (green), and VE-cadherin (white, junctions). Arrow indicates flow direction. Insets show junctions. Scale bar, 20 μm. (**E**) Quantification of cell alignment shown in (**D**). n=59 cells (NT) and 73 cells (SUN1 KD) compiled from three replicates. (**F**) Quantification of streptavidin area shown in (**D**). n=15 ROIs (NT) and 15 ROIS (SUN1 KD) compiled from three replicates. (**G**) Representative images of HUVEC with indicated siRNAs showing adherens following EDTA washout. Endothelial cells were stained for DAPI (cyan, DNA) and VE-cadherin (white, junctions). Insets show junctions. Scale bar, 20 μm. (**H**) Quantification of VE-cadherin line scans at 20, 40, and 60 min post EDTA washout in (**G**). 20 min: n=31 junctions (NT) and 23 junctions (SUN1 KD); 40 min: n=49 junctions (NT) and 33 junctions (SUN1 KD); 60 min: n=33 junctions (NT) and 33 junctions (SUN1 KD) compiled from three replicates. ns, not significant; ****, p<0.0001 by Student's two-tailed unpaired t-test. For C, error bars represent standard deviation. For E, F, and H, boxes represent the upper quartile, lower quartile, and median; whiskers represent the minimum and maximum values.

The online version of this article includes the following source data and figure supplement(s) for figure 4:

**Figure supplement 1.** SUN1 regulates endothelial junction integrity.

**Figure supplement 1—source data 1.** Western blot showing four experimental replicates.

junctions (*Figure 4—figure supplement 1F–G*). Thus, SUN1 is not required to form endothelial cell-cell junctions but is necessary for proper junction maturation and stabilization.

## SUN1 regulates microtubule localization and dynamics in endothelial cells absent effects on gene transcription

We next considered how SUN1 that resides in the nuclear membrane regulates cell behaviors at the cell periphery. SUN1 may affect cell junctions directly via the cytoskeleton, or SUN1 may indirectly affect junctions downstream of gene expression regulation. Because SUN1 is reported to regulate gene transcription and RNA export in other cell types (*Li et al., 2017*; *May and Carroll, 2018*), we asked whether endothelial cell transcriptional profiles were altered by SUN1 depletion. To our surprise, RNASeq analysis of HUVEC under both static and flow conditions revealed essentially no significant changes in RNA profiles except for *SUN1* itself (*Table 1*, *Figure 5—figure supplement 1*), while in the same experiment we documented extensive expression changes in both up and down-regulated genes between control HUVEC in static vs. laminar flow conditions, as we and others have shown (*Conway et al., 2010*; *Liu et al., 2021*; *Maurya et al., 2021*; *Ruter et al., 2021*). These findings are similar to a re-analysis of HeLa cell data (data not shown; *Li et al., 2017*) that revealed few significant changes in gene expression following SUN1 depletion. Thus, although the LINC complex is important for nuclear communication that can affect gene expression, SUN1 is not required for this communication in endothelial cells.

We hypothesized that SUN1 directly regulates cell junction stability via the cytoskeleton. SUN1 has a functional relationship with the microtubule cytoskeleton (*Zhu et al., 2017*), and microtubule dynamics regulate endothelial cell-cell junctions (*Sehrawat et al., 2008*; *Sehrawat et al., 2011*; *Komarova et al., 2012*). Thus, we asked whether the observed adherens junction defects following SUN1 depletion resulted from changes in the microtubule cytoskeleton. Microtubule depolymerization via nocodazole treatment phenocopied SUN1 KD and destabilized adherens junctions in control monolayers but did not exacerbate the junction defects seen with SUN1 KD (*Figure 5A–B*). These findings suggest that endothelial cell junction defects are downstream of microtubule perturbations induced by SUN1 depletion.

We next investigated how SUN1 affects microtubule localization in endothelial cell monolayers and found that significantly fewer microtubules reached the cell periphery or surrounded the nucleus in

**Table 1.** SUN1 depletion does not alter endothelial gene expression.

| Condition | # Upregulated DEG | # Downregulated DEG |
|---|---|---|
| *NT_FLOW vs. NT_STAT* | 1323 | 1109 |
| *SUN1_STAT vs. NT_STAT* | 0 | 1 |
| *SUN1_FLOW vs. NT_FLOW* | 1 | 1 |

Bold numbers indicate that single downregulated gene was *SUN1*.
Abbreviation: NT, non-targeting; STAT, static; DEG, differentially expressed genes.

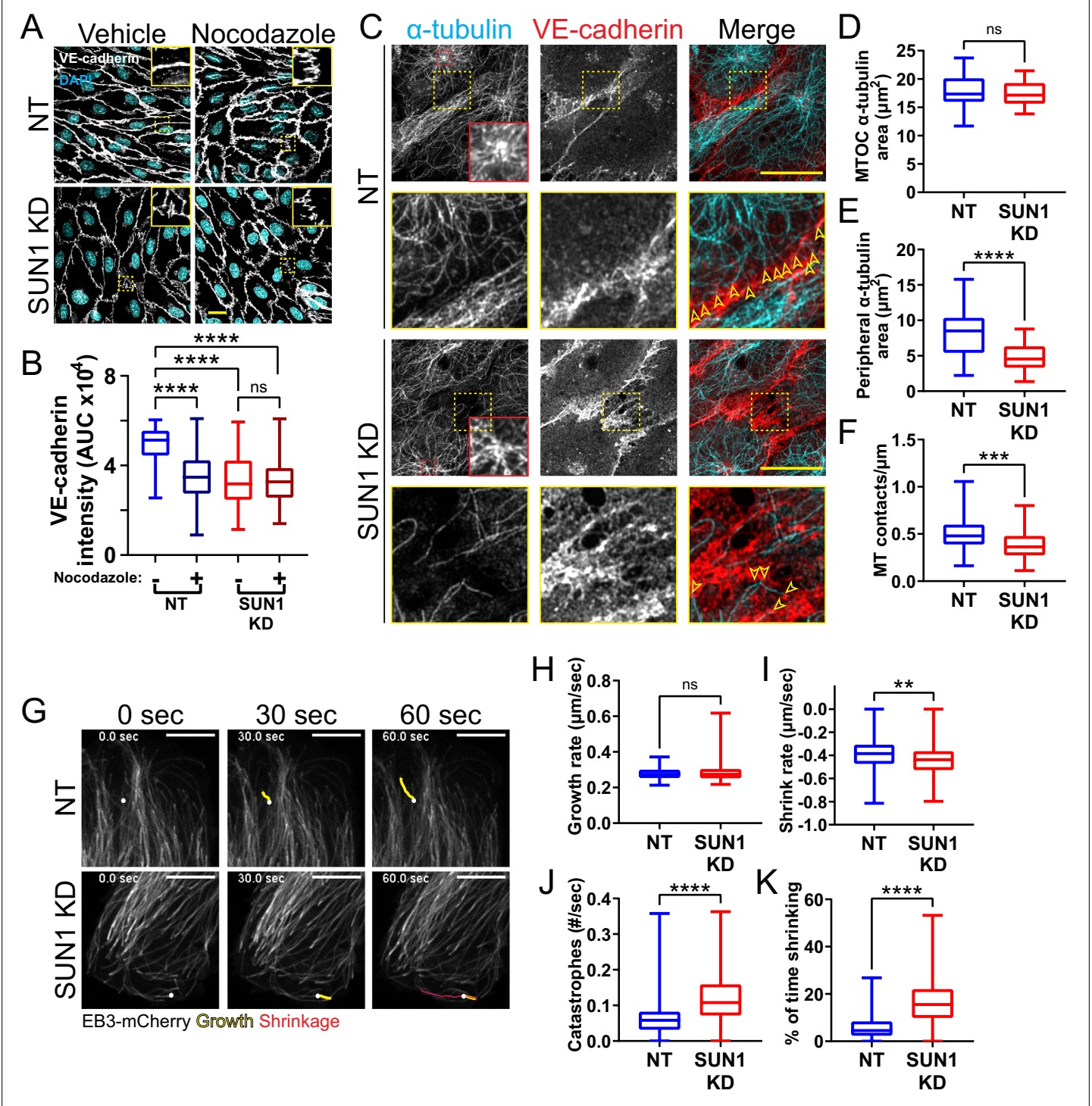

**Figure 5.** SUN1 regulates microtubule localization and dynamics in endothelial cells. (**A**) Representative images of human umbilical vein endothelial cells (HUVEC) with indicated siRNAs and indicated treatments. Endothelial cells were stained for DAPI (cyan, DNA) and VE-cadherin (white, junctions). Insets show junctions. Scale bar, 20 μm. (**B**) Quantification of VE-cadherin line scans for treatments shown in (**A**). n=106 junctions (non-targeting [NT], vehicle), 101 junctions (NT, Nocodazole), 105 junctions (SUN1 knockdown [KD], vehicle), and 96 junctions (SUN1 KD, Nocodazole) compiled from three replicates. ns, not significant; ****, p<0.0001 by two-way ANOVA with Tukey's multiple comparisons test. (**C**) Representative images of HUVEC with indicated siRNAs. Endothelial cells were stained for α-tubulin (cyan, microtubules) and VE-cadherin (red, junctions). Red insets show α-tubulin at the MTOC (microtubule organizing center), yellow insets show α-tubulin contacts at junctions. Arrows denote contact sites. Scale bar, 20 μm. (**D**) Quantification of α-tubulin area at the MTOC shown in (**C**). n=19 cells (NT) and 10 cells (SUN1 KD) compiled from three replicates. ns, not significant

*Figure 5 continued on next page*

*Figure 5 continued*

by Student's two-tailed unpaired *t*-test. (**E**) Quantification of peripheral α-tubulin area shown in (**C**). n=39 cells (NT) and 46 cells (SUN1 KD) compiled from three replicates. ****, p<0.0001 by Student's two-tailed unpaired *t*-test. (**F**) Quantification of contacts between α-tubulin and VE-cadherin shown in (**C**). n=75 junctions (NT) and 48 junctions (SUN1 KD) compiled from three replicates. ***, p<0.001 by Student's two-tailed unpaired *t*-test. (**G**) Stills from *Video 5* and *Video 6* showing microtubule growth in EB3-mCherry labeled HUVEC. White dot indicates start of track. Yellow line indicates growth, red line indicates shrinkage. Scale bar, 10 μm. (**H**) Quantification of microtubule growth rate from EB3-mCherry microtubule tracking. N=120 microtubules (12 cells, NT) and 117 microtubules (12 cells, SUN1 KD) compiled from two replicates. ns, not significant by Student's two-tailed unpaired *t*-test. (**I**) Quantification of microtubule shrink rate from EB3-mCherry microtubule tracking. n=120 microtubules (12 cells, NT) and 117 microtubules (12 cells, SUN1 KD) compiled from two replicates. **, p<0.01 by Student's two-tailed unpaired *t*-test. (**J**) Quantification of catastrophe rate from EB3-mCherry microtubule tracking. n=120 microtubules (12 cells, NT) and 117 microtubules (12 cells, SUN1 KD) compiled from two replicates. ****, p<0.0001 by Student's two-tailed unpaired *t*-test. (**K**) Quantification of percent of time spent shrinking from EB3-mCherry microtubule tracking. n=120 microtubules (12 cells, NT) and 117 microtubules (12 cells, SUN1 KD) compiled from two replicates. ****, p<0.0001 by Student's two-tailed unpaired *t*-test. For all graphs, boxes represent the upper quartile, lower quartile, and median; whiskers represent the minimum and maximum values.

The online version of this article includes the following figure supplement(s) for figure 5:

**Figure supplement 1.** SUN1 does not affect transcription in endothelial cells.

**Figure supplement 2.** SUN1 regulates the microtubule cytoskeleton in endothelial cells.

SUN1-depleted cells (*Figure 5C and E*, *Figure 5—figure supplement 2A–B*). The changes in peripheral microtubule localization were accompanied by significantly fewer microtubule-junction contacts, while α-tubulin levels around the MTOC were not affected (*Figure 5C–D and F*). Microtubule dynamics were assessed via tip tracking using mCherry-labeled tip protein EB3, which decorated the microtubule lattice and concentrated at growing microtubule tips (*Figure 5G*). This labeling pattern can occur with EB overexpression but does not affect growth rate (*Komarova et al., 2005*), so the decorated lattice was used to assess microtubule catastrophe and shrinkage. Microtubules in SUN1-depleted cells had increased shrinkage rates (a more negative value) coupled with increased catastrophe rate and time spent shrinking (*Figure 5G–K*, *Video 5* and *Video 6*), consistent with elevated microtubule depolymerization and impaired microtubule dynamics downstream of SUN1 loss, while the microtubule growth rate was unchanged. Taken together, these results indicate that loss of SUN1 impairs microtubule localization and dynamics, and these changes associate with destabilized endothelial cell junctions.

## SUN1 regulates endothelial cell contractility

Microtubule dynamics communicate with the actin cytoskeleton to regulate cell-cell junctions (*Verin et al., 2001*; *Birukova et al., 2004*; *Birukova et al., 2006*), and cellular changes in actomyosin contractility contribute to junction destabilization (*Rauzi et al., 2010*; *Huveneers et al., 2012*). We hypothesized that actomyosin mis-regulation induced by SUN1 depletion contributes to endothelial cell junction destabilization and found that SUN1 depletion led to ectopic stress fibers, distinct radial actin bundles at the periphery, and increased phosphorylated myosin light chain (ppMLC), consistent with increased actomyosin contractility (*Figure 6—figure supplement 1A–B*). Pharmacological

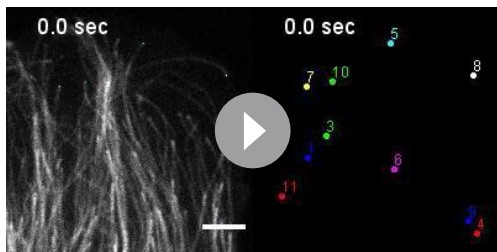

**Video 5.** Control endothelial cells have normal microtubule dynamics. Movie taken for 120 s in human umbilical vein endothelial cells (HUVEC) with non-targeting (NT) siRNA labeled with EB3-mCherry. Quantified microtubule tracks are indicated. Scale bar, 5 μm. Frames acquired every 500 ms.
https://elifesciences.org/articles/83652/figures#video5

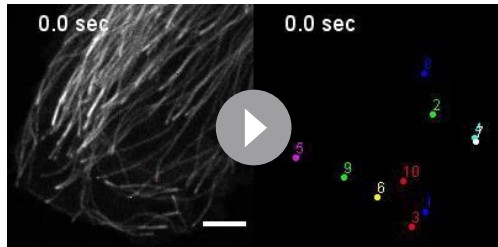

**Video 6.** Loss of SUN1 in endothelial cells leads to impaired microtubule dynamics. Movie taken for 120 s in human umbilical vein endothelial cells (HUVEC) with SUN1 siRNA labeled with EB3-mCherry. Quantified microtubule tracks are indicated. Scale bar, 5 μm. Frames acquired every 500 ms.
https://elifesciences.org/articles/83652/figures#video6

blockade of myosin-II ATPase rescued the destabilized cell junctions, ectopic stress fibers, and radial actin structures seen with SUN1 depletion (*Figure 6A–B*, *Figure 6—figure supplement 1C*), suggesting that SUN1 depletion increases contractility. In contrast, thrombin treatment that induces contractility produced disorganized junctions in controls that phenocopied SUN1 depletion but did not contribute to further junction destabilization in SUN1 KD cells, indicating that SUN1 loss induces a maximal contractile state (*Figure 6C–D*). Taken together, these data indicate that SUN1 loss results in hypercontractility and destabilized cell-cell junctions.

## SUN1 affects endothelial junctions through microtubule-associated Rho GEF-H1

We next wanted to better understand the link between microtubule dynamics, actomyosin contractility, and junction regulation downstream of SUN1. SUN1 silencing increases RhoA activity in HeLa cells (*Thakar et al., 2017*), and pharmacological inhibition of Rho kinase (ROCK) signaling in SUN1-depleted endothelial cells rescued the destabilized cell-cell junctions and prevented the radial actin structures (*Figure 6—figure supplement 2A–C*). RhoA signaling and barrier function in endothelial cells is regulated by the microtubule-associated RhoGEF, GEF-H1, which is inactive while bound to microtubules and peaks in activation following microtubule depolymerization and release into the cytosol (*Krendel et al., 2002*; *Birukova et al., 2006*; *Birkenfeld et al., 2008*; *Azoitei et al., 2019*). We hypothesized that the impaired microtubule dynamics and Rho-dependent hypercontractility observed following SUN1 depletion were linked via GEF-H1. We first examined Rho activation via pulldowns and found increased activated Rho with SUN1 silencing that was abrogated when GEF-H1 was also depleted (*Figure 6—figure supplement 2D*, *Figure 6—figure supplement 3A*), suggesting that GEF-H1 signaling is upstream of elevated RhoA activation in SUN1-depleted cells. Consistent with this idea, GEF-H1 strongly localized to peripheral microtubules in control cells but was significantly less localized and more diffuse in the cytosol in SUN1-depleted endothelial cells (*Figure 6E–F*), suggesting its release from microtubules. Furthermore, depletion of GEF-H1 in endothelial cells that were also depleted for SUN1 rescued the destabilized cell-cell junctions and loss of junction integrity observed with SUN1 KD alone (*Figure 6G–J*), showing that GEF-H1 is required to transmit the effects of SUN1 depletion to endothelial cell junctions. Interestingly, depletion of GEF-H1 was not sufficient to rescue the loss of microtubule density at the periphery in SUN1-depleted cells (*Figure 6—figure supplement 3B–C*), indicating that GEF-H1 acts downstream of changes in the microtubule cytoskeleton in SUN1-depleted cells. Thus, nuclear SUN1 normally promotes microtubule-GEFH1 interactions to regulate actomyosin contractility and endothelial cell-cell junction stability via RhoA, providing a novel linkage from the LINC complex to cell junctions via the microtubule cytoskeleton.

## SUN1 exerts its effects on endothelial junctions through nesprin-1

Since SUN1 binds nesprins to interact with the cytoskeleton, we considered whether SUN1-nesprin interactions were involved in SUN1 regulation of microtubule dynamics and endothelial cell junction stability. The KASH protein nesprin-1 modulates tight junction protein localization under laminar shear stress (*Yang et al., 2020*) and regulates microtubule dynamics at the nucleus in muscle syncytia (*Gimpel et al., 2017*). Co-depletion of SUN1 with nesprin-1 in endothelial cells rescued the effects of SUN1 depletion on junction morphology and functional destabilization measured by matrix biotin labeling (*Figure 7A and D*, *Figure 7—figure supplement 1A*), while nesprin-1 depletion alone did not affect these parameters (*Figure 7—figure supplement 1B–C*). This rescue extended to the cytoskeleton, as co-depletion of SUN1 and nesprin-1 rescued the decreased peripheral microtubule density and microtubule-GEF-H1 contacts seen with SUN1 depletion (*Figure 7B–C and E–F*), while nesprin-1 depletion alone did not have an effect (*Figure 7—figure supplement 1D–G*). Thus, the LINC complex protein nesprin-1 is required to transmit the effects of SUN1 depletion to endothelial cell junctions.

## Discussion

The nucleus compartmentalizes and organizes genetic material, but how the nucleus directly communicates with other organelles and the cytoskeleton to regulate cell behaviors is poorly understood. Here, we show for the first time that the nuclear LINC complex protein SUN1 regulates angiogenic sprouting and endothelial cell-cell junction integrity via long-distance regulation in vivo, and that

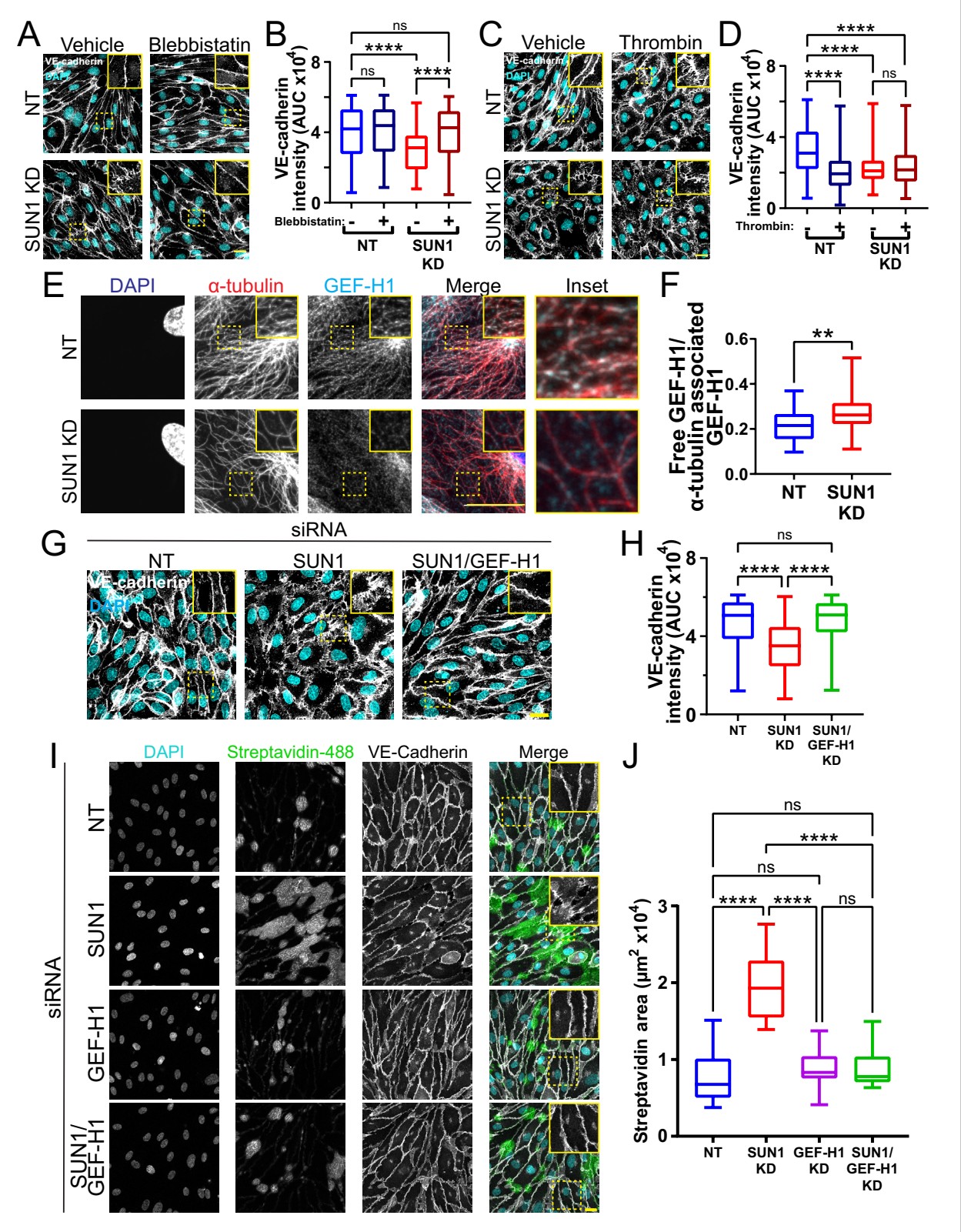

**Figure 6.** SUN1 regulates endothelial cell contractility and exerts its effects on junctions through the microtubule-associated GEF-H1. (**A**) Representative images of human umbilical vein endothelial cells (HUVEC) with indicated siRNAs and indicated treatments. Endothelial cells were stained for DAPI (cyan, DNA) and VE-cadherin (white, junctions). Insets show junctions. Scale bar, 20 μm. (**B**) Quantification of VE-cadherin line scans for treatments shown in (**A**). n=159 junctions (non-targeting [NT], vehicle), 154 junctions (NT, blebbistatin), 151 junctions (SUN1 knockdown [KD], vehicle),

*Figure 6 continued on next page*

*Figure 6 continued*

and 149 junctions (SUN1 KD, blebbistatin) compiled from three replicates. ns, not significant; ****, p<0.0001 by two-way ANOVA with Tukey's multiple comparisons test. (**C**) Representative images of HUVEC with indicated siRNAs and indicated treatments. Endothelial cells were stained for DAPI (cyan, DNA) and VE-cadherin (white, junctions). Insets show junctions. Scale bar, 20 μm. (**D**) Quantification of VE-cadherin line scans for treatments shown in (**C**). n=75 junctions (NT, vehicle), 70 junctions (NT, thrombin), 71 junctions (SUN1 KD, vehicle), and 73 junctions (SUN1 KD, thrombin) compiled from three replicates. ns, not significant; ****, p<0.0001 by two-way ANOVA with Tukey's multiple comparisons test. (**E**) Representative images of HUVEC with indicated siRNAs. Endothelial cells were stained for DAPI (blue, DNA), α-tubulin (red, microtubules), and GEF-H1 (cyan). Insets show α-tubulin and GEF-H1 colocalization. Scale bar, 20 μm. (**F**) Quantification of free GEF-H1 normalized to α-tubulin associated GEF-H1 shown in (**E**). n=30 cells (NT) and 30 cells (SUN1 KD) compiled from three replicates. **, p<0.01 by Student's two-tailed unpaired *t*-test. (**G**) Representative images of HUVEC with indicated siRNAs and indicated treatments. Endothelial cells were stained for DAPI (cyan, DNA) and VE-cadherin (white, junctions). Insets show junctions. Scale bar, 20 μm. (**H**) Quantification of VE-cadherin line scans from KD shown in (**G**). n=169 junctions (NT), 166 junctions (SUN1 KD), 170 junctions (SUN1/GEF-H1 KD) compiled from three replicates. ns, not significant; ****, p<0.0001 by one-way ANOVA with Tukey's multiple comparisons test. (**I**) Representative images of HUVEC with indicated siRNAs cultured on biotinylated fibronectin and treated with streptavidin upon confluence. Endothelial cells were stained for DAPI (cyan, DNA), streptavidin (green), and VE-cadherin (white, junctions). Insets show junctions. Scale bar, 20 μm. (**J**) Quantification of streptavidin area shown in (**I**). n=15 ROIs (NT), 15 ROIs (SUN1 KD), 15 ROIs (GEF-H1 KD), and 15 ROIs (SUN1/GEF-H1 KD) compiled from three replicates. ns, not significant; ****, p<0.0001 by one-way ANOVA with Tukey's multiple comparisons test. For all graphs, boxes represent the upper quartile, lower quartile, and median; whiskers represent the minimum and maximum values.

The online version of this article includes the following source data and figure supplement(s) for figure 6:

**Figure supplement 1.** SUN1 regulates the actin cytoskeleton in endothelial cells.

**Figure supplement 2.** SUN1 acts through RhoA to affect junctions.

**Figure supplement 2—source data 1.** Western blots showing representative replicate.

**Figure supplement 3.** GEF-H1 signaling is downstream of SUN1 and microtubules.

**Figure supplement 3—source data 1.** Western blot showing representative replicate.

these effects go through the microtubule cytoskeleton to regulate microtubule dynamics and acto-myosin contractility. Our data is consistent with a model in which SUN1 stabilizes peripheral micro-tubules that coordinate activity of the microtubule-associated Rho exchange factor GEF-H1. GEF-H1 becomes activated to stimulate Rho signaling upon release from microtubules, and we posit that peripheral microtubules normally regulate local GEF-H1 activity to maintain appropriate microtubule-junction interactions and actomyosin contractility in endothelial cells, leading to cell-cell junctions that remodel to support angiogenic sprouting and maintain vessel barrier integrity (*Figure 8*). Loss of SUN1 reduces peripheral microtubules, resulting in GEF-H1 over-activation, elevated RhoA signaling, and increased contractility, leading to destabilized endothelial cell adherens junctions that impair blood vessel formation and function in fish and mice. The LINC complex is further implicated in endo-thelial junction regulation by our finding that the KASH protein nesprin-1 is required for the effects of SUN1 loss on microtubules, GEF-H1, and junction integrity. Thus, we describe a specific role for SUN1 as a critical mediator of communication between the endothelial cell nucleus and the cell periphery via microtubule regulation of GEF-H1. We investigated the role of SUN1 in endothelial cell and vascular function; however, the molecular machinery whereby SUN1 regulates cell junctions is not specific to endothelial cells. Thus, SUN1 likely influences cell-cell junctions via similar mechanisms in other cell and tissue types.

We show that the LINC complex protein SUN1 is required for endothelial adherens junction stabi-lization and proper blood vessel formation and function in vivo. The adherens junctions of expanding retinal vessels are destabilized in mice lacking endothelial *Sun1*, and this cellular phenotype is accom-panied by increased leak, reduced radial expansion, and increased network density, indicating that destabilized junctions contribute to the vessel network perturbations. Interestingly, mice with global knockout of *Sun1* are viable (*Lei et al., 2009*; *Zhang et al., 2009*) and zebrafish injected with MO against *sun1b* resemble controls by 72 hpf (data not shown), suggesting that the developmental disruption of the vascular network does not perdure. This may be due to partial redundancy with SUN2 (*Lei et al., 2009*; *Zhang et al., 2009*) preventing more profound vascular effects. However, the significant vascular defects in vessels lacking *Sun1* reveal non-redundant functions for SUN1 in vascular development. We show that depletion of SUN1 leads to impaired junction morphology and integrity as measured by increased biotin matrix labeling and decreased impedance in cultured cells and increased dextran leak in the postnatal retina. Because junction morphology and integrity are integral to maintenance of the endothelial barrier, these findings suggest that SUN1 depletion leads

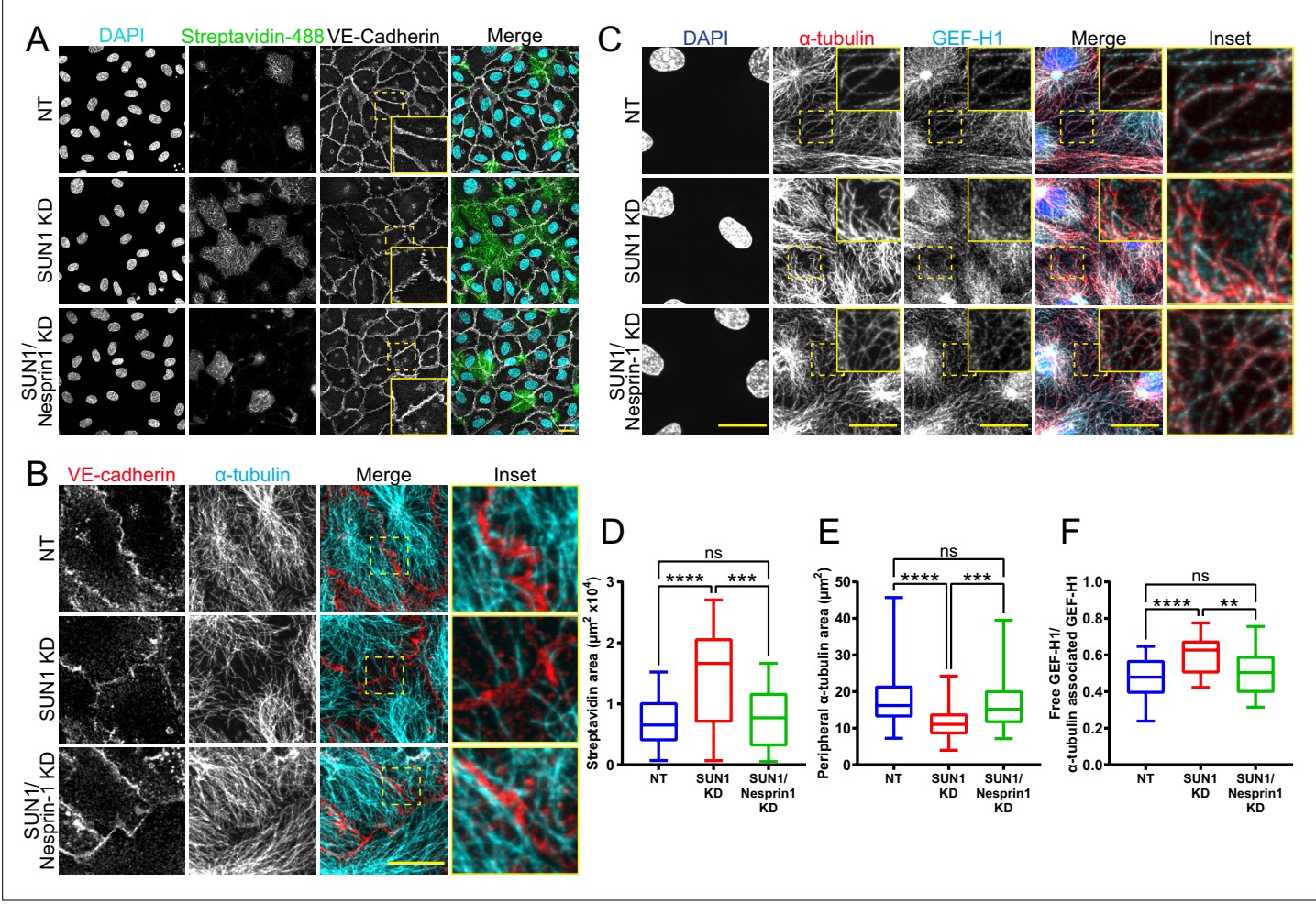

**Figure 7.** SUN1 regulates endothelial cell junctions through nesprin-1. (**A**) Representative images of human umbilical vein endothelial cells (HUVEC) with indicated siRNAs cultured on biotinylated fibronectin and treated with streptavidin upon confluence. Endothelial cells were stained for DAPI (cyan, DNA), streptavidin (green), and VE-cadherin (white, junctions). Insets show junctions. Scale bar, 20 μm. (**B**) Representative images of HUVEC with indicated siRNAs. Endothelial cells were stained for α-tubulin (cyan, microtubules) and VE-cadherin (red, junctions). Insets show α-tubulin contacts at junctions. Scale bar, 20 μm. (**C**) Representative images of HUVEC with indicated siRNAs. Endothelial cells were stained for DAPI (blue, DNA), α-tubulin (red, microtubules), and GEF-H1 (cyan). Insets show α-tubulin and GEF-H1 colocalization. Scale bar, 20 μm. (**D**) Quantification of streptavidin area shown in (**A**). n=22 ROIs (non-targeting [NT]), 22 ROIs (SUN1 knockdown [KD]), and 22 ROIs (SUN1/nesprin-1 KD) compiled from four replicates. ns, not significant; ***, p<0.001; ****, p<0.0001 by one-way ANOVA with Tukey's multiple comparisons test. (**E**) Quantification of peripheral α-tubulin area shown in (**B**). n=52 cells (NT), 52 cells (SUN1 KD), and 52 cells (SUN1/nesprin-1 KD) compiled from three replicates. ns, not significant; ***, p<0.001; ****, p<0.0001 by one-way ANOVA with Tukey's multiple comparisons test. (**F**) Quantification of free GEF-H1 normalized to α-tubulin associated GEF-H1 shown in (**C**). n=32 cells (NT), 32 cells (SUN1 KD), and 32 cells (SUN1/nesprin-1 KD) compiled from three replicates. ns, not significant; **, p<0.01; ****, p<0.0001 by one- way ANOVA with Tukey's multiple comparisons test. For all graphs, boxes represent the upper quartile, lower quartile, and median; whiskers represent the minimum and maximum values.

The online version of this article includes the following figure supplement(s) for figure 7:

**Figure supplement 1.** Loss of nesprin-1 alone does not impact endothelial junctions or microtubules.

to impaired barrier function. Mutations in other genes that affect endothelial cell junction integrity, such as *Smad6*, *Pi3kca*, and *Yap/Taz*, also perturb retinal angiogenesis (*Angulo-Urarte et al., 2018*; *Neto et al., 2018*; *Wylie et al., 2018*). Live image analysis of active vessel sprouting showed that SUN1 regulates filopodia dynamics and anastomosis in zebrafish and sprout dynamics in mammalian endothelial cell sprouts, and these vessels also had abnormal junctions in the absence of SUN1. Altered sprout dynamics are found in other scenarios where VE-cadherin is absent or abnormal such as Wnt inhibition, loss of VE-cadherin, PI3-kinase inhibition, and excess centrosomes (*Sauteur et al., 2014*; *Sauteur et al., 2017*; *Kushner et al., 2016*; *Angulo-Urarte et al., 2018*; *Hübner et al., 2018*;

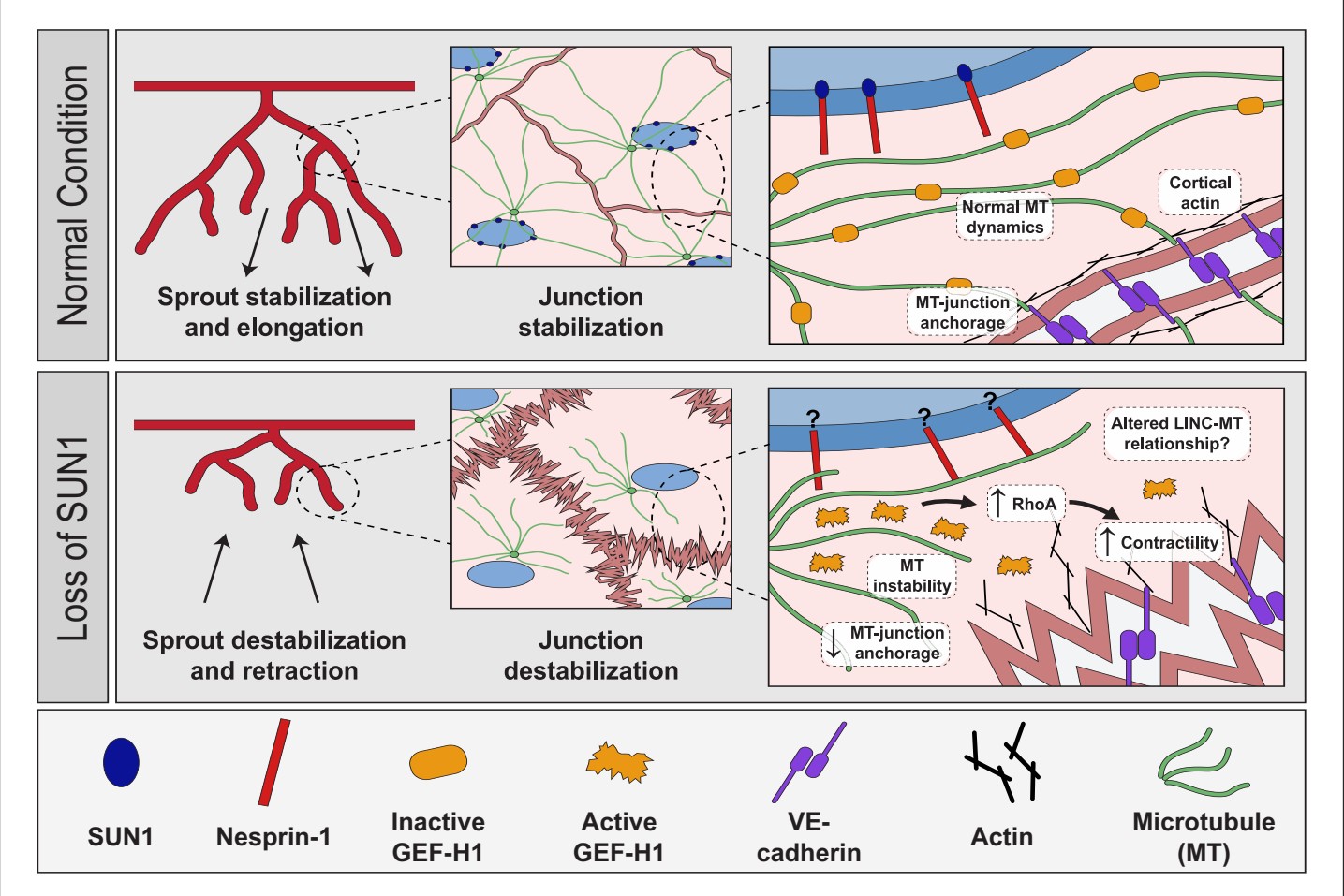

**Figure 8.** Proposed role of SUN1 in angiogenic sprouting and endothelial cell junction stabilization. Model describing proposed role of SUN1 in angiogenic sprouting and endothelial cell junction stabilization.

*Buglak et al., 2020*). The LINC complex regulates endothelial cell aggregation into tube-like structures in Matrigel (*King et al., 2014*; *Denis et al., 2021*), consistent with our findings that highlight a central role for SUN1 in blood vessel formation in vivo. Unlike genes encoding components of junctions and signaling effectors, SUN1 functions at significant cellular distances from cell junctions.

How does SUN1 regulate cell-cell junctions from a distance? The importance of the LINC complex in transducing signals from the cell periphery and outside the cell to the nucleus to affect nuclear envelope properties and gene transcription is well studied (*Carley et al., 2022*), but how information goes from the nucleus to the cell periphery is less well understood. The LINC complex is important in nucleus-to-cytoplasm communication, but most published studies assess manipulations that affect the LINC complex globally (*Zhang et al., 2009*; *Graham et al., 2018*; *Carley et al., 2021*; *Denis et al., 2021*), and how individual components contribute is not well understood. *Ueda et al., 2022*, found that SUN1 regulates focal adhesion maturation in non-endothelial cells in vitro via effects on the actin cytoskeleton (*Ueda et al., 2022*), and we found that SUN1 regulates endothelial cell-cell junctions through microtubules, indicating that SUN1 is important for signaling from the nucleus to the cell periphery by regulating cytoskeletal organization at several levels. Since LINC complex manipulations are reported to affect focal adhesions and cell migration (*King et al., 2014*; *Denis et al., 2021*; *Ueda et al., 2022*), it is likely that vessel dysfunction following SUN1 manipulation is complex. However, SUN1 is thought to be important in microtubule-associated LINC complex functions (*Zhu et al., 2017*), and disruption of microtubules or their dynamics destabilizes adherens junctions in both endothelial and non-endothelial cells (*Komarova et al., 2012*; *Vasileva and Citi, 2018*). Our data show that SUN1 depletion impairs both microtubule dynamics and microtubule localization. Specifically, elevated rates

of microtubule catastrophe and shrinkage absent changes in growth rate likely account for reduced peripheral microtubule density and microtubule-junction contacts. Endothelial junction destabilization was phenocopied by microtubule depolymerization, consistent with our model that the influence of SUN1 on microtubule dynamics and localization is important for the stabilization of cell junctions.

Here, we show that SUN1 regulates cell junctions via the microtubule-regulated Rho activator GEF-H1. GEF-H1 specifically modulates RhoA signaling at endothelial cell adherens junctions to influence VE-cadherin internalization (*Juettner et al., 2019*). Microtubule depolymerization releases GEF-H1 to activate RhoA signaling and elevate actomyosin contractility (*Krendel et al., 2002*; *Birkenfeld et al., 2008*; *Azoitei et al., 2019*) in non-endothelial cells, while direct manipulation of GEF-H1 via depletion or blockade attenuates agonist-induced endothelial barrier dysfunction (*Birukova et al., 2006*). Our work revealed that peripheral GEF-H1 localization was more diffuse following endothelial SUN1 silencing, suggesting its activation with SUN1 loss, and Rho-kinase inhibition or GEF-H1 depletion rescued SUN1 depletion-induced RhoA activation and destabilization of endothelial cell junctions. These findings support that altered microtubule dynamics downstream of SUN1 depletion promote the release and activation of peripheral GEF-H1, and this activation destabilizes cell junctions and likely affects vascular barrier function.

Thus, SUN1 regulation of microtubule dynamics is linked to its regulation of junction stability, although exactly how SUN1 influences microtubule dynamics and function is unclear. SUN1 resides in the nuclear envelope and alters gene transcription in other cell types (*Li et al., 2017*; *May and Carroll, 2018*). Our transcriptional analysis of endothelial cells under flow did not reveal significant transcriptome changes with SUN1 depletion, indicating that SUN1 regulation of endothelial junctions occurs via its role in LINC complex interactions with the cytoskeleton. Nesprin-1, a KASH protein that functions in LINC complex-cytoskeletal interactions, is shown here to mediate the effects of SUN1 loss on endothelial cell junctions, including junction morphology and function and peripheral microtubule density, suggesting that SUN1 normally sequesters nesprin-1 to prevent formation of ectopic complexes. SUN1 may normally prevent abnormal or unstable nesprin-1 LINC complexes that promote microtubule depolymerization, and SUN1 loss allows these complexes to form. SUN1 may compete for nesprin-1 binding with other nuclear envelope proteins that interact with nesprin-1, such as SUN2 or nesprin-3 (*Stewart-Hutchinson et al., 2008*; *Taranum et al., 2012*; *Yang et al., 2020*). This idea is consistent with the finding that SUN1 antagonizes SUN2-based LINC complexes that promote RhoA activity in HeLa cells, although microtubule localization changes were not reported (*Thakar et al., 2017*).

Our finding that SUN1 regulates endothelial cell junction integrity and blood vessel sprouting has implications for diseases associated with aging, as vascular defects underlie most cardiovascular disease. Children with Hutchinson-Gilford progeria syndrome (HGPS) have a mutation in the *LMNA* gene encoding lamin A/C, resulting in accumulation of an abnormal lamin protein called progerin; these patients age rapidly and die in their early to mid-teens from severe atherosclerosis (*De Sandre-Giovannoli et al., 2003*; *Eriksson et al., 2003*; *Olive et al., 2010*). Progerin has increased SUN1 affinity that leads to SUN1 accumulation in HGPS patient cells (*Haque et al., 2010*; *Chen et al., 2012*; *Chen et al., 2014*; *Chang et al., 2019*). Endothelial cells also accumulate SUN1 in HGPS mouse models (*Osmanagic-Myers et al., 2019*), and loss of *Sun1* partially rescues progeria phenotypes in mouse models and patient cells (*Chen et al., 2012*; *Chang et al., 2019*). Thus, nuclear membrane perturbations affecting SUN1 cause disease, and here we find that nuclear SUN1 regulates microtubules to affect both the microtubule and actin cytoskeletons. These effects are transmitted to endothelial cell-cell junctions far from the site of SUN1 localization to influence endothelial cell behaviors, blood vessel sprouting, and barrier function.

## Materials and methods

### Microscopy

Unless otherwise stated, all imaging was performed as follows: confocal images were acquired with an Olympus confocal laser scanning microscope and camera (Fluoview FV3000, IX83) using 405, 488, 561, and 640 nm lasers and a UPlanSApo 40× silicone-immersion objective (NA 1.25), UPlanSApo 60× oil-immersion objective (NA 1.40), or UPlanSApo 100× oil-immersion objective (NA 1.40). Imaging was performed at RT for fixed samples. Images were acquired with the Olympus Fluoview FV31S-SW

software and all image analysis, including Z-stack compression, was performed in Fiji (*Linkert et al., 2010*; *Schindelin et al., 2012*). Any adjustments to brightness and contrast were performed evenly for images in an experiment.

## Mice

All experimental mouse (*Mus musculus*) procedures performed in this study were reviewed and approved by the University of North Carolina Chapel Hill Institutional Animal Care and Use Committee (IACUC) (PHS Animal Welfare Assurance Number D16-00256 [A3410-010]; AAALAC Accreditation #329). We thank the Centre d'ImmunoPhenomique-Ciphe for providing the $Sun1^{tm1a}$ mutant mouse line (Allele: B6NJ;B6N-Sun1$^{tm1a(EUCOMM)Wtsi}$/CipheOrl), INFRAFRONTEIR/EMMA (https://www.infrafrontier.eu, REF 25414328), and the Institut de Transgenose from which the mouse line was distributed (EM:09532). Associated primary phenotypic information may be found at https://www.mousephenotype.org/. FlpO-B6N-Albino (Rosa26-FlpO/+) mice were obtained from the UNC Animal Models Core. *Cdh5-CreERT2* mice were generated by Dr Ralf Adams (*Sörensen et al., 2009*) and obtained from Cancer Research UK. The $Sun1^{tm1a}$ allele was identified via genomic PCR to amplify the LacZ insertion (Forward: 5'- ACTATCCCGACCGCCTTACT-3'; Reverse: 5'- TAGCGGCTGATGTTGAACTG -3'). The $Sun1^{fl}$ allele was generated by breeding $Sun1^{tm1a}$ mice with FlpO-B6N-Albino (Rosa26-FlpO/+) mice to excise the *lacZ* insertion. The $Sun1^{fl}$ allele was identified via genomic PCR using the following primers (Forward: 5'- GCTCTCTGAAACATGGCTGA-3'; Reverse: 5'- ATCCGGGGTGTT TGGATTAT-3'). $Sun1^{fl}$ mice were bred to *Cdh5-CreERT2* mice to generate $Sun1^{fl/fl}$;*Cdh5-CreERT2* pups for endothelial-selective and temporally controlled deletion of exon 4 of the *Sun1* gene. The excised *Sun1* allele was identified via genomic PCR on lung tissue using the following primers (Forward: 5'- CTTTTGGGCTGCTCTGTTGT-3'; Reverse: 5'- ATCCGGGGTGTTTGGATTAT-3'). PCR genotyping for FlpO and Cdh5-CreERT2 mice was performed with the following primers (FlpO: Forward: 5'-TGAGC TTCGACATCGTGAAC –3'; Reverse: 5'-TCAGCATCTTCTTGCTGTGG-3') (Cdh5-CreERT2: Forward: 5'- TCCTGATGGTGCCTATCCTC-3'; Reverse: 5'- CCTGTTTTGCACGTTCACCG-3'). Induction of Cre was performed via IP injection of pups at P1, P2, and P3 with 50 µl of 1 mg/ml tamoxifen (T5648, Sigma) dissolved in sunflower seed oil (S5007, Sigma). Littermates lacking either *Cdh5-CreERT2* or the $Sun1^{fl}$ allele were used as controls.

## Mouse retinas

Tamoxifen-injected mice were sacrificed at P7, eyes were collected, fixed in 4% PFA for 1 hr at RT, then dissected and stored at 4°C in PBS for up to 2 weeks (*Chong et al., 2017*). Retinas were permeabilized in 0.5% Triton X-100 (T8787, Sigma) for 1 hr at RT, blocked for 1 hr at RT in blocking solution (0.3% Triton X-100, 0.2% BSA (A4503, Sigma)), and 5% goat serum (005-000-121, Jackson ImmunoResearch), then incubated with VE-cadherin antibody (anti-mouseCD144, 1:100, 550548, BD Pharmingen) or ERG antibody (1:500, ab196149, Abcam) in blocking solution overnight at 4°C. Samples were washed 3×, then incubated with Isolectin B4 AlexaFluor 488 (1:100, I21411, Thermo Fisher) and goat anti-rat AlexaFluor 647 (1:500, A21247, Life Technologies) for 1 hr at RT. Retinas were mounted with Prolong Diamond Antifade mounting medium (P36961, Life Technologies) and sealed with nail polish. Images were obtained using either a UPlanSApo 10× air objective (NA 0.40) or UPlanSApo 40× silicone-immersion objective (NA 1.25). Percent radial expansion was calculated by dividing the distance from the retina center to the vascular front by the distance from the retina center to the edge of the tissue with four measurements per retina. Vascular density was measured by imaging a 350 µm × 350 µm ROI at the vascular edge. Fiji was used to threshold images, and the vessel area was normalized to the area of the ROI (n=4 ROI/retina, chosen at 2 arteries and 2 veins). Junctions were measured by taking the ratio of the mean intensity of the junction and the mean intensity of the area immediately adjacent to the junction. Mean intensity was measured via line scans in Fiji. Sixteen junctions/retina were measured.

## Retina blood vessel permeability

Tamoxifen-injected mice were anesthetized at P7 in isoflurane for 5 min. The abdomen was opened, and the diaphragm was cut. One-hundred µl of 5 mg/ml 10 kDa Dextran-Texas Red (D1863, Invitrogen) in PBS was injected into the left ventricle of the heart. Eyes were immediately collected and fixed in 4% PFA for 1 hr at RT, then dissected and stained as described above. Leak was determined

by making a mask of the vessel area using the isolectin channel, then assessing the dextran signal outside the vessel.

## 3D sprouting angiogenesis assay

The 3D sprouting angiogenesis assay was performed as previously described (*Nakatsu and Hughes, 2008*; *Nesmith et al., 2017*). 48 hr following siRNA KD, HUVEC were coated onto cytodex 3 micro-carrier beads (17048501, GE Healthcare Life Sciences) and embedded in a fibrin matrix by combining 7 µl of 50 U/ml thrombin (T7201-500UN, Sigma) with 500 µl of 2.2 mg/ml fibrinogen (820224, Fisher) in a 24-well glass-bottomed plate (662892, Grenier Bio). The matrix was incubated for 20 min at RT followed by 20 min at 37°C to allow the matrix to solidify. EGM-2 was then added to each well along with 200 µl of normal human lung fibroblasts (CC2512, Lonza) at a concentration of $2×10^5$ cells/ml. At day 7 of sprouting, fibroblasts were removed via trypsin treatment (5X-trypsin for 3 min at 37°C), and samples were fixed in 4% PFA for 15 min at RT. 0.5% Triton X-100 in DBPS was added to the wells, and incubation was overnight at 4°C. After rinsing 3× in DPBS, samples were blocked (5% goat serum [005-000-121, Jackson ImmunoResearch], 1% BSA [A4503, Sigma], and 0.3% Triton X-100 [T8787, Sigma]) overnight at 4°C. Samples were rinsed 3× in DPBS then anti-VE-cadherin antibody (1:1000, 2500S, Cell Signaling) in blocking solution was added for 24 hr at 4°C. Samples were rinsed 3× 10 min in 0.5% Tween 20 then washed overnight at 4°C in 0.5% Tween. Samples were rinsed 3× in DPBS, then DAPI (0.3 µM, 10236276001, Sigma) and AlexaFluor488 Phalloidin (1:50, A12379, Life Technologies) in blocking solution were added to the wells, and incubation was overnight at 4°C prior to rinsing 3× in DPBS. For whole bead analysis, images were acquired in the Z-plane using a UPlanSApo 20× oil-immersion objective (NA 0.58) and processed in Fiji. Average sprout length was measured by tracing each sprout from base (bead) to tip, then averaging lengths per bead. Branching was measured by counting total branch points and normalizing to total sprout length per bead using the AnalyzeSkeleton plugin (*Arganda-Carreras et al., 2010*). For junctions, images were acquired in the Z-plane using a UPlanSApo 60× oil-immersion objective (NA 1.40). Junctions were measured by taking the ratio of the mean intensity of the junction and the mean intensity of the area immediately adjacent to the junction. Mean intensity was measured via line scans in Fiji.

Live imaging on HUVEC sprouts was performed between days 4 and 6.5 of sprouting. Images were acquired on an Olympus VivaView Incubator Fluorescence Microscope with a UPLSAPO 20× objective (NA 0.75) and ×0.5 magnification changer (final magnification ×20) with a Hamamatsu Orca R2 cooled CCD camera at 30 min intervals for 60 hr at 37°C. Images were acquired using the MetaMorph imaging software and analyzed in Fiji. Sprouts were considered to have 'retracted' if they regressed toward the bead for at least three imaging frames (1.5 hr).

## Zebrafish

All experimental zebrafish (*Danio rerio*) procedures performed in this study were reviewed and approved by the University of North Carolina Chapel Hill Institutional Animal Care and Use Committee (IACUC) (PHS Animal Welfare Assurance Number D16-00256 [A3410-010]; AAALAC Accreditation #329). Animals were housed in an AAALAC-accredited facility in compliance with the *Guide for the Care and Use of Laboratory Animals* as detailed on protocols.io (dx.doi.org/10.17504/protocols.io.bg3jjykn). *Tg(fli:LifeAct-GFP)* was a gift from Dr Wiebke Herzog. *sun1b^sa33109* mutant fish were obtained from the Zebrafish International Resource Center (ZIRC). For genotyping, the target region of the *sun1b* gene was amplified via genomic PCR using the following primers (Forward: 5'-GGCTGCGTCAGACTCCATTA-3'; Reverse: 5'-TTGAGTTAAACCCAGCGCCT-3'). The amplicon was then sequenced by Sanger sequencing (GENEWIZ) using the forward primer. Morphant fish were obtained by randomly injecting 2.5–5 ng of non-targeting (NT) (5'-CCTCTTACCTCAGTTACAATTTATA-3', GeneTools, LLC) or *sun1b* (5'-CGCAGTTTGACCATCAGTTTCTACA-3', GeneTools, LLC) MO into *Tg(fli:LifeAct-GFP)* embryos at the one-cell stage. Fish were grown in E3 medium at 28.5°C to 33–34 hpf.

## Zebrafish imaging

Dechorionated embryos were incubated in ice-cold 4% PFA at 4°C overnight or RT for 2 hr. Embryos were permeabilized in 0.5% Triton X-100 in PBST (DPBS+0.1% Tween 20) for 1 hr at RT then blocked (PBST+0.5% Triton X-100+1% BSA+5% goat serum+0.01% sodium azide) for 2 hr at RT. Anti-ZO1 primary antibody (1:500, 33-9100, Thermo Fisher) was added overnight at 4°C.

Embryos were rinsed in PBST overnight at 4°C. Goat anti-mouse AlexaFluor647 secondary antibody (1:1000, A21236, Life Technologies) was added overnight at 4°C. Embryos were washed 3× in PBST for 30 min then overnight in PBST at 4°C. Embryos were rinsed in PBS and a fine probe was used to de-yolk and a small blade to separate the trunk from the cephalic region. Samples were mounted using Prolong Diamond Antifade mounting medium (P36961, Life Technology) and the coverslip was sealed with petroleum jelly. Imaging was at RT using a UPlanSApo 20× oil-immersion objective (NA 0.58) or a UPlanSApo 60× oil-immersion objective (NA 1.40) with an additional 3× magnification, for a total magnification of 180×. Filopodia length was measured by drawing a line from the filopodia base to the tip. Filopodia number was measured by counting the number of filopodia and normalizing to the total vessel length. Filopodia in at least three areas per fish were measured. Junctions were analyzed by drawing a line along the junction then normalizing to the shortest distance between the two ends of the junction. At least three junctions were measured per fish.

For live imaging of zebrafish, fish were dechorionated and then anesthetized with 1× Tricaine in E3 for 5 min. Fish were embedded in 0.5% agarose in 1× Tricaine-E3 medium on a glass-bottomed plate in a stage-top incubator (TOKAI HIT, WSKM) at 28.5°C. Images were acquired using a UPlanSApo 40× air objective (NA 0.95) every 15 min for 10–15 hr. ISVs that did not reach the DLAV or connected at non-consistent intervals were considered to have a missing or aberrant DLAV connection if at least one ISV posterior to the scored ISV made a normal connection.

## Cell culture

Commercially available primary HUVEC (C2519A, Lonza, mixed sex) were cultured in EBM-2 (CC-3162, Lonza) supplemented with the Endothelial Growth Medium (EGM-2) bullet kit (CC-3162, Lonza) and 1× antibiotic-antimycotic (Gibco). Commercially available primary normal human lung fibroblasts (CC2512, Lonza, mixed sex) were cultured in DMEM (Gibco) with 10% fetal bovine serum (FBS) and 1× antibiotic-antimycotic (Gibco). All cells were maintained at 37°C and 5% $CO_2$. For contractility inhibition experiments, HUVEC were treated with 10 µM (-) Blebbistatin (B0560-1MG, Sigma) for 15 min at 37°C or 10 µM Y-27632 (10187-694, VWR) for 30 min at 37°C then immediately fixed in 4% PFA. For induction of contractility, HUVEC were treated with 0.5 U/ml thrombin (T7201-500UN, Sigma) for 10 min at 37°C. For microtubule depolymerization, HUVEC were treated with 10 µM nocodazole (M1404, Sigma) for 20 min at 37°C then immediately fixed in methanol.

## siRNA KD

HUVEC were transfected with non-targeting siRNA (NT, #4390847, Life Technologies), SUN1 siRNA #1 (439240, #s23630, Life Technologies), SUN1 siRNA #2 (439240, #s23629, Life Technologies), GEF-H1 siRNA (439240, #s17546, Life Technologies), and/or nesprin-1 siRNA (M-014039-02-0005, Dharmacon) using Lipofectamine 2000 (11668027, Invitrogen) or Lipofectamine 3000 (L3000015, Thermo Fisher). siRNA at 0.48 µM in Opti-MEM (31985-070, Gibco) and a 1:20 dilution of Lipofectamine in Opti-MEM were incubated separately at RT for 5 min, then combined and incubated at RT for 15 min. HUVEC were transfected at ~80% confluency with siRNA at 37°C for 24 hr, then recovered in EGM-2 for an additional 24 hr. HUVEC were seeded onto glass chamber slides coated with 5 µg/ml fibronectin (F2006-2MG, Sigma) for experiments.

## RNA sequencing and analysis

RNA was extracted using TRIzol (15596018, Invitrogen) from three biological replicates (independent experiments) of HUVEC under static or homeostatic laminar flow (15 dyn/cm², 72 hr) conditions, and KAPA mRNA HyperPrep Kit (7961901001, Roche) was used to prepare stranded libraries for sequencing (NovaSeq S1). $2–3×10^7$ 50 bp paired-end reads per sample were obtained and mapped to human genome GRCh38 downloaded from https://support.10xgenomics.com/single-cell-gene-expression/software/pipelines/latest/advanced/references with STAR using default settings (*Dobin et al., 2013*). Mapping rate was over 80% for all samples, and gene expression was determined with Htseq-count using the union mode (https://htseq.readthedocs.io/en/master/) (*Putri et al., 2022*). Differential expression analysis was performed with DESeq2 (*Love et al., 2014*) using default settings in R, and lists of differentially expressed genes were obtained (p adjusted <0.1).

## Immunofluorescence staining

For experiments visualizing microtubules, HUVEC were fixed in ice-cold methanol for 10 min at 4°C. For all other experiments, HUVEC were fixed with 4% PFA for 10 min at RT and permeabilized with 0.1% Triton X-100 (T8787, Sigma) for 10 min at RT. Fixed HUVEC were blocked for 1 hr at RT in blocking solution (5% FBS, 2× antibiotic-antimycotic [Gibco], 0.1% sodium azide [s2002-100G, Sigma] in DPBS). Cells were incubated in primary antibody overnight at 4°C, then washed 3× for 5 min in DPBS. Secondary antibody and DRAQ7 (1:1000, ab109202, Abcam), DAPI (0.3 µM, 10236276001, Sigma), and/or AlexaFluor488 Phalloidin (1:100, A12379, Life Technologies) were added for 1 hr at RT followed by 3× washes for 10 min each in DPBS. Slides were mounted with coverslips using Prolong Diamond Antifade mounting medium (P36961, Life Technologies) and sealed with nail polish. Primary and secondary antibodies were diluted in blocking solution. The following primary antibodies were used: anti-VE-cadherin (1:500, 2500S, Cell Signaling), anti-SUN1 (1:500, ab124770, Abcam), anti-Ki67 (1:500, ab15580, Abcam), anti-phospho-myosin light chain 2 (Thr18/Ser19) (1:500, 3674S, Cell Signaling), anti-alpha-tubulin (1:500, 3873S, Cell Signaling), anti-GEF-H1 (1:500, ab155785, Abcam), and anti-SYNE1 (1:500, HPA019113, Atlas Antibodies). The following secondary antibodies from Life Technologies were used: goat anti-mouse AlexaFluor 488 (1:500, A11029), goat anti-rabbit AlexaFluor 594 (1:500, A11037), goat anti-mouse 647 (1:500, A21236), and goat anti-rabbit 647 (1:500, A21245).

## Western blotting

Cells were scraped into RIPA buffer with protease/phosphatase inhibitor (5872S, Cell Signaling) then centrifuged at 13,000 rpm at 4°C for 20 min. Lysate was reduced in sample loading buffer and dithiothreitol (R0861, Thermo Fisher) and boiled for 10 min at 100°C. Samples were stored at –20°C until use. Ten µg of sample were run on a 10% stain-free polyacrylamide gel (161-0183, Bio-Rad) then transferred onto a PVDF membrane on ice for 1.5 hr. Membranes were blocked in OneBlock (20-313, Prometheus) for 1 hr at RT then washed 3× in PBST. Anti-GEF-H1 (1:1000, ab155785, Abcam), anti-VE-cadherin (1:14,000, 2500S, Cell Signaling), or anti-GAPDH (1:5000, 97166S, Cell Signaling) was added overnight at 4°C. Membranes were washed 3× in PBST then donkey anti-rabbit HRP secondary antibody (1:10,000, A16035, Thermo Fisher) was added for 1 hr at RT. Immobilon Forte HRP Substrate (WBLUF0100, MilliporeSigma) was added for 30 s. Blots were exposed for 8 s.

## EdU labeling

HUVEC were labeled with EdU using the Click-It EdU Kit 488 (Invitrogen, C10337) and fixed according to the manufacturer's instructions. Cells positive for EdU labeling were counted and compared to total cell number to obtain percent positive.

## Junction analysis

Endothelial cell adherens junctions were quantified in monolayers using Fiji to generate 15 µm line scans of VE-cadherin signal parallel to the cell junctions. VE-cadherin signal was integrated to obtain the area under the curve. Linear junctions with consistent VE-cadherin signal thus had a large area under the curve, while more serrated junctions had reduced area under the curve (*Figure 4—figure supplement 1E*). Measurements were performed on at least 9 cells per field of view, with 3–6 fields of view per condition.

## Real time cell analysis

Barrier function was assessed using the xCELLigence Real-Time Cell Analyzer (RTCA, Acea Biosciences/Roche Applied Science) to measure electrical impedance across HUVEC monolayers seeded onto microelectrodes. HUVEC were seeded to confluency on the microelectrodes of the E-plate (E-plate 16, Roche Applied Science). Electrical impedance readings were taken every 2 min for 5 hr. The percent change in cell index was obtained at the 5 hr timepoint using the following formula: (Cell Index$_{SUN1}$-Cell Index$_{NT}$)/ABS(Cell Index$_{NT}$).

## Flow experiments

Flow experiments were performed using an Ibidi pump system (10902, Ibidi) as described (*Ruter et al., 2021*). HUVEC were seeded onto fibronectin coated Ibidi slides either µ-Slide I Luer I 0.4 mm (80176, Ibidi) or µ-Slide Y-shaped (80126, Ibidi) in flow medium (EBM-2 with 10% FBS and

1× Antibiotic-antimycotic). HUVEC were exposed to laminar shear stress for 30 min at 5 dyn/cm$^2$, followed by 30 min at 10 dyn/cm$^2$, and finally for 72 hr at 15 dyn/cm$^2$. Alignment was measured by taking the ratio of the longitudinal axis to the transverse axis relative to the flow vector. At least 10 cells were measured per condition per experiment. Vascular permeability in vitro was determined using the biotin matrix labeling assay as described below.

## Biotin matrix labeling assay

Labeling of biotinylated matrix was assessed as described (*Dubrovskyi et al., 2013*). Briefly, fibronectin was biotinylated by incubating 0.1 mg/ml fibronectin with 0.5 mM EZ-Link Sulfo-NHS-LC-Biotin (A39257, Thermo Fisher) for 30 min at RT. Glass chamber slides were coated with 5 µg/ml biotinylated-fibronectin and HUVEC were seeded on top. At confluency, HUVEC were treated with 25 µg/ml Streptavidin-488 (S11223, Invitrogen) for 3 min then immediately fixed. For quantification, Fiji was used to threshold the streptavidin signal, and the streptavidin area was measured and normalized for total area for at least three fields of view per experiment.

## Junction reformation assay

The EDTA junction reformation assay was performed as previously described (*Wright et al., 2015*). Briefly, HUVEC were treated with 3 mM EDTA (EDS-100G, Sigma-Aldrich) for 1 hr at 37°C. EDTA was then washed out 3× with DPBS, incubation at 37°C in EGM-2 was continued, and cells were fixed at 0, 20, 40, and 60 min intervals.

## VE-cadherin internalization

VE-cadherin internalization was performed as described (*Wylie et al., 2018*). Briefly, HUVEC were plated on 5 µg/ml fibronectin and grown to confluency. After overnight serum starvation (Opti-MEM [31985–070, Gibco] supplemented with 1% FBS [F2442, Sigma], and 1× antibiotic-antimycotic [Gibco]), cells were washed with pre-chilled PBS+ (14040182, Thermo Fisher) on ice at 4°C, then incubated in ice-cold blocking solution (EBM-2 [CC-3162, Lonza] supplemented with 0.5% BSA [A4503, Sigma]) for 30 min at 4°C. HUVEC were then incubated with VE-cadherin BV6 antibody (1:100, ALX-803-305C100, Enzo) in blocking solution for 2 hr on ice at 4°C. Following VE-cadherin labeling, cells were washed with PBS+ then incubated in pre-warmed internalization medium (EBM-2) at 37°C for 1 hr. Finally, HUVEC were incubated in acid wash (0.5 M NaCl/0.2 M acetic acid) for 4 min at 4°C to remove remaining labeled VE-cadherin on the cell surface, then washed with PBS+ and fixed. For quantification, internalized VE-cadherin area was measured in Fiji, then normalized to total cell area for at least 9 cells per experiment.

## Microtubule analysis

For nuclear microtubule analysis, an ROI was drawn in Fiji over the nucleus. Fiji was used to threshold α-tubulin signal, and the area of the signal was measured within the ROI. This was performed on 8 cells per field of view, with 3–6 fields per condition per experiment. For high-resolution microtubule analysis, high-resolution confocal images were acquired with a Zeiss 880 Confocal with AiryScan FAST microscope with GaAsP detector and camera (Zeiss) using a Plan-Apo 63× oil-immersion objective (NA 1.40) and 488 and 561 nm lasers. Imaging was performed at RT, and images were acquired with the Zeiss 880 software and with the AiryScan detector. Images were then processed with AiryScan. All image analysis, including Z-stack compression, was performed in Fiji. For microtubule density, an ROI was drawn at the MTOC and the cell periphery. Fiji was used to threshold α-tubulin signal, and the area of the signal was measured within the ROI. This was performed on at least 10 cells per condition per experiment. For junction analysis, the number of contacts between α-tubulin and VE-cadherin was counted and normalized to the junction length. At least 15 junctions were measured per condition per experiment.

## Microtubule tip tracking

HUVEC were infected with an EB3-mCherry Lentivirus (*Kushner et al., 2014*) for 24 hr to visualize microtubule comets. Following infection, HUVEC were incubated with siRNAs for NT or SUN1. For live imaging, cells were incubated at 37°C in a stage-top incubator (TOKAI HIT, WSKM). Images were acquired on an Andor XD spinning disk confocal microscope based on a CSU-X1 Yokogawa head with an Andor iXon 897 EM-CCD camera. A 561 nm laser and FF01-607/36 emission filter were used. Images were acquired with

a 470 ms exposure and 32 ms readout time for 2 min (240 frames) using a UPlanSApo 60× silicone-oil-immersion objective (NA 1.30) and Metamorph software. Microtubules at the cell periphery were tracked using the Manual Tracking plugin in Fiji and were tracked for at least 60 frames (30 s). Track information was acquired from the x and y coordinates using a custom algorithm in Visual Basic in Excel provided by Dr Dan Buster at the University of Arizona. Ten microtubule tracks were measured per cell.

### GEF-H1 analysis

For GEF-H1 localization analysis, an ROI was drawn at the periphery of the cell, and the α-tubulin signal was used to create a mask. Mean signal intensity was then measured for GEF-H1 within the α-tubulin mask and outside of it, and a ratio was taken. At least 9 cells were analyzed per condition per experiment.

### RhoA activity assay

Cells were placed on ice, washed with ice-cold PBS and lysed in 10 mM $MgCl_2$, 500 mM NaCl, 50 mM Tris, pH 7.6, 1% Triton X-100, 0.1% SDS, 0.5% deoxycholate, 1 mM phenylmethylsulfonyl fluoride (PMSF), 10 µg/ml aprotinin and leupeptin. Lysates were clarified by centrifugation and rotated with 50 µg of glutathione-Sepharose-bound glutathione *S*-transferase-RBD (Rhotekin-binding domain) for 25 min at 4°C. Beads were then washed 3× in 50 mM Tris, pH 7.6, 10 mM $MgCl_2$, 150 mM NaCl, 1% Triton X-100, 1 mM PMSF, and 10 µg/ml aprotinin and leupeptin. Released proteins and reserved input control were subjected to western blot analysis.

Samples were resolved on polyacrylamide gels in the presence of SDS. Resolved gels were transferred onto nitrocellulose membranes, blocked with 5% BSA in Tris-buffered saline (25 mM Tris, pH 7.6, 150 mM NaCl) plus 0.1% Tween-20 (TBST) and incubated with anti-RhoA primary antibody (1:1000, 2117, Cell Signaling) overnight at 4°C with gentle rocking. Blots were washed extensively in TBST, then incubated with species-appropriate HRP-conjugated secondary antibody (111-035-144, Jackson ImmunoResearch) for 1 hr at RT. Blots were again washed in TBST and developed using an enhanced chemiluminescent reagent (Thermo Fisher Scientific) and X-ray film.

### Statistics

Student's two-tailed unpaired *t*-test was used to determine statistical significance in experiments with two groups and one-way ANOVA with Tukey's multiple comparisons test was used in experiments with three or more groups. For thrombin, blebbistatin, Y-27632, and nocodazole experiments, two-way ANOVA with Tukey's multiple comparisons test was used to determine statistical significance. $X^2$ was used for categorical data. For box and whisker plots, boxes represent the upper quartile, lower quartile, and median; whiskers represent the minimum and maximum values. Statistical tests and graphs were made using the Prism 9 software (GraphPad Software).

## Acknowledgements

We thank Michelle Altemara and staff at the Zebrafish Aquaculture Core and Kaitlyn Quigley for fish room support, Aaron Friedman and Caroline Crater for mouse room support, and Dr. Yosuke Mukouyama, Dr. Nick Buglak, Dr. Keith Burridge and Bautch Lab members for critical discussion and feedback, and Dr. Nick Buglak for artwork. We also thank Dr. Dan Buster for the software used in microtubule tip tracking experiments and Dr. Angelika Noegel for sharing RNASeq data with us. Airy Scan imaging was performed at the UNC Hooker Imaging Core Facility and microtubule tip tracking imaging was performed with Dr. Pablo Ariel at the UNC Microscopy Services Laboratory, supported in part by P30 CA016086 Cancer Center Core Support Grant to the UNC Lineberger Comprehensive Cancer Center. This work was supported by grants from the National Institutes of Health (R35 HL139950 to VLB), the Integrated Vascular Biology Training Grant (5T32HL069768-17, DBB), and an American Heart Association Predoctoral Fellowship (19PRE34380887 to DBB).

## Additional information

#### Competing interests

Victoria L Bautch: Reviewing editor, eLife. The other authors declare that no competing interests exist.

## Funding

| Funder | Grant reference number | Author |
|---|---|---|
| National Institutes of Health | R35 HL139950 | Victoria L Bautch |
| National Institutes of Health | 5T32HL069768-17 | Danielle B Buglak |
| American Heart Association | 19PRE34380887 | Danielle B Buglak |

The funders had no role in study design, data collection and interpretation, or the decision to submit the work for publication.

## Author contributions

Danielle B Buglak, Conceptualization, Data curation, Formal analysis, Investigation, Methodology, Writing – original draft, Writing – review and editing; Pauline Bougaran, Molly R Kulikauskas, Ariel L Gold, Allison P Marvin, Andrew Burciu, Morgan Oatley, Shea N Ricketts, Karina Kinghorn, Bryan N Johnson, Investigation, Methodology; Ziqing Liu, Data curation, Formal analysis, Methodology, Writing – review and editing; Elizabeth Monaghan-Benson, Data curation, Investigation, Methodology; Natalie T Tanke, Investigation, Methodology, Writing – review and editing; Celia E Shiau, Christophe Guilluy, Supervision, Methodology; Stephen Rogers, Conceptualization, Methodology; Victoria L Bautch, Conceptualization, Resources, Supervision, Funding acquisition, Methodology, Writing – original draft, Project administration, Writing – review and editing

## Author ORCIDs

Danielle B Buglak  http://orcid.org/0000-0002-8702-4018
Molly R Kulikauskas  http://orcid.org/0000-0001-5186-9740
Morgan Oatley  http://orcid.org/0000-0001-9465-6585
Victoria L Bautch  http://orcid.org/0000-0003-2135-5153

## Ethics

This study was performed in strict accordance with the recommendations in the Guide for the Care and Use of Laboratory Animals of the National Institutes of Health. All animals were maintained and handled according to approved institutional animal care and use committee (IACUC) protocols of the University of North Carolina at Chapel Hill (PHS Animal Welfare Assurance Number D16-00256 (A3410-010); AAALAC Accreditation #329). The protocol was approved by the Committee on the Ethics of Animal Experiments of the University of North Carolina.

## Decision letter and Author response

Decision letter https://doi.org/10.7554/eLife.83652.sa1
Author response https://doi.org/10.7554/eLife.83652.sa2

# Additional files

## Supplementary files
• MDAR checklist

## Data availability

Bulk RNA Sequencing data has been deposited in GEO under accession code GSE213099.

The following dataset was generated:

| Author(s) | Year | Dataset title | Dataset URL | Database and Identifier |
|---|---|---|---|---|
| Buglak D, Gold A, Kulikauskas K, Ricketts S, Marvin A, Burciu A, Tanke N | 2023 | BULK RNA SEQ OF HUVEC - NON-TARGETING AND SUN1, STATIC AND FLOW | https://www.ncbi.nlm.nih.gov/geo/query/acc.cgi?acc=GSE213099 | NCBI Gene Expression Omnibus, GSE213099 |

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

# Appendix 1

### Appendix 1—key resources table

| Reagent type (species) or resource | Designation | Source or reference | Identifiers | Additional information |
|---|---|---|---|---|
| Gene (*Mus musculus*) | *Sun1* | Ensembl | Ensembl_ID: ENSMUSG00000036817 | |
| Gene (*Homo sapiens*) | *SUN1* | Ensembl | Ensemble_ID: ENSG00000164828 | |
| Gene (*Danio rerio*) | *sun1b* | ZFIN | ZFIN_ID:ZDB -GENE-050522–551 | |
| Strain, strain background (*M. musculus*) | B6NJ;B6N-Sun1^tm1a(EUCOMM)/Wtsi/ CipheOrl | European Mouse Mutant Archive (EMMA) | EMMA_ID:EM:09532 | |
| Strain, strain background (*M. musculus*) | FlpO-B6N-Albino (Rosa26-FlpO/+) | Other | | UNC Animal Models Core |
| Strain, strain background (*M. musculus*) | Tg(Cdh5-cre/ERT2)1Rha | *Sörensen et al., 2009* (PMID: 19144989) | | Dr Ralf Adams |
| Strain,strain background (*D. rerio*) | Tg(fli:LifeAct-GFP) | Other | | Dr Wiebke Herzog |
| Strain,strain background (*D. rerio*) | sun1b^sa33109 | Zebrafish International Resource Center (ZIRC) | Cat#:ZL12625.02 | |
| Cell line (*Homo sapiens*) | Human Umbilical Vein Endothelial Cells (HUVEC) | Lonza | Cat#:C2519A | Human primary endothelial cells, mixed sex |
| Cell line (*H. sapiens*) | Normal Human Lung Fibroblasts (NHLF) | Lonza | Cat#:CC2512 | Human primary lung fibroblast cells, mixed sex |
| Transfected construct (*H. sapiens*) | Non-targeting (NT) siRNA | Life Technologies | Cat#:4390847 | Silencer select |
| Transfected construct (*H. sapiens*) | SUN1 siRNA #1 | Life Technologies | Cat#:439240; #s23630 | Silencer select |
| Transfected construct (*H. sapiens*) | SUN1 siRNA #2 | Life Technologies | Cat#:439240; #s23629 | Silencer select |
| Transfected construct (*H. sapiens*) | GEF-H1 siRNA | Life Technologies | Cat#:439240; #s17546 | Silencer select |
| Transfected construct (*H. sapiens*) | Nesprin-1 siRNA | Dharmacon | Cat#: M-014039-02-0005 | SMARTpool |
| Transfected construct (*H. sapiens*) | EB3-mCherry | *Kushner et al., 2014* (PMID: 25049273) | | Lentiviral construct |
| Antibody | Anti-mouseCD144 (rat monoclonal) | BD Pharmingen | Cat#:550548 | Primary antibody, detects VE-cadherin in mouse tissue IF mouse (1:100) |
| Antibody | Anti-ERG (rabbit monoclonal) | Abcam | Cat#:ab196149 | Primary antibody conjugated to AlexaFluor647, detects nuclei in endothelial cells IF mouse (1:500) |

*Appendix 1 Continued on next page*

*Appendix 1 Continued*

| Reagent type (species) or resource | Designation | Source or reference | Identifiers | Additional information |
|---|---|---|---|---|
| Antibody | Anti-VE-cadherin (rabbit monoclonal) | Cell Signaling | Cat#:2500S | Primary antibody, detects human VE-cadherin IF cells 3D(1:1000) IF cells 2D (1:500) Western (1:14000) |
| Antibody | Anti-ZO1 (mouse monoclonal) | Thermo Fisher | Cat#:33-9100 | Primary antibody, detects ZO1 in zebrafish IF zebrafish (1:500) |
| Antibody | Anti-SUN1 (rabbit monoclonal) | Abcam | Cat#:ab124770 | Primary antibody, detects human SUN1 IF cells (1:500) |
| Antibody | Anti-Ki67 (rabbit polyclonal) | Abcam | Cat#:ab15580 | Primary antibody IF cells (1:500) |
| Antibody | Anti-phospho-myosin light chain 2 (Thr18/Ser19) (rabbit polyclonal) | Cell Signaling | Cat#:3674S | Primary antibody IF cells (1:500) |
| Antibody | Anti-alpha-tubulin (mouse monoclonal) | Cell Signaling | Cat#:3873 S | Primary antibody IF cells (1:500) |
| Antibody | Anti-GEF-H1 (rabbit polyclonal) | Abcam | Cat#:ab155785 | Primary antibody IF cells (1:500) Western (1:1000) |
| Antibody | Anti-SYNE1 (rabbit polyclonal) | Atlas Antibodies | Cat#:HPA019113 | Primary antibody IF cells (1:500) |
| Antibody | Anti-GAPDH (mouse monoclonal) | Cell Signaling | Cat#:97166S | Primary antibody Western (1:5000) |
| Antibody | Anti-VE-cadherin BV6 (mouse monoclonal) | Enzo | Cat#:ALX-803-305C100 | Primary antibody, detects the extracellular region of VE-cadherin IF cells (1:100) |
| Antibody | Anti-RhoA (rabbit monoclonal) | Cell Signaling | Cat#:2117 | Primary antibody Western (1:1000) |
| Antibody | Goat anti-mouse AlexaFluor 488 (goat polyclonal) | Life Technologies | Cat#:A11029 | Secondary antibody IF cells (1:500) |
| Antibody | Goat anti-rabbit AlexaFluor 594 (goat polyclonal) | Life Technologies | Cat#:A11037 | Secondary antibody IF cells (1:500) |
| Antibody | Goat anti-rat AlexaFluor 647 (goat polyclonal) | Life Technologies | Cat#:A21247 | Secondary antibody IF mouse (1:500) |
| Antibody | Goat anti-mouse AlexaFluor 647 (goat polyclonal) | Life Technologies | Cat#:A21236 | Secondary antibody IF zebrafish (1:1000) IF cells (1:500) |
| Antibody | Goat anti-rabbit AlexaFluor 647 (goat polyclonal) | Life Technologies | Cat#:A21245 | Secondary antibody IF cells (1:500) |
| Antibody | Donkey anti-rabbit HRP (donkey polyclonal) | Thermo Fisher | Cat#:A16035 | Secondary antibody Western (1:10,000) |
| Antibody | Goat anti-rabbit HRP (goat polyclonal) | Jackson ImmunoResearch | Cat#:111-035-144 | Secondary antibody Western |
| Sequence-based reagent | LacZ_F | This paper | PCR primers | ACTATCCCGACCGCCTTACT |
| Sequence-based reagent | LacZ_R | This paper | PCR primers | TAGCGGCTGATGTTGAACTG |
| Sequence-based reagent | Sun1$^{fl}$_F | This paper | PCR primers | GCTCTCTGAAACATGGCTGA |
| Sequence-based reagent | Sun1$^{fl}$_R | This paper | PCR primers | ATCCGGGGTGTTTGGATTAT |

*Appendix 1 Continued on next page*

*Appendix 1 Continued*

| Reagent type (species) or resource | Designation | Source or reference | Identifiers | Additional information |
|---|---|---|---|---|
| Sequence-based reagent | Sun1excised_F | This paper | PCR primers | CTTTTGGGCTGCTCTGTTGT |
| Sequence-based reagent | Sun1excised_R | This paper | PCR primers | ATCCGGGGTGTTTGGATTAT |
| Sequence-based reagent | FlpO_F | This paper | PCR primers | TGAGCTTCGACATCGTGAAC |
| Sequence-based reagent | FlpO_R | This paper | PCR primers | TCAGCATCTTCTTGCTGTGG |
| Sequence-based reagent | Cdh5Cre_F | This paper | PCR primers | TCCTGATGGTGCCTATCCTC |
| Sequence-based reagent | Cdh5Cre_R | This paper | PCR primers | CCTGTTTTGCACGTTCACCG |
| Sequence-based reagent | sun1b_F | This paper | PCR primers | GGCTGCGTCAGACTCCATTA |
| Sequence-based reagent | sun1b_R | This paper | PCR primers | TTGAGTTAAACCCAGCGCCT |
| Sequence-based reagent | Non-targeting (NT) morpholino (MO) | This paper | | CCTCTTACCTCAGTTACAAT TTATA |
| Sequence-based reagent | *sun1b* morpholino (MO) | This paper | | CGCAGTTTGACCATCAGTTT CTACA |
| Peptide, recombinant protein | Isolectin B4 AlexaFluor 488 | Thermo Fisher | Cat#:I21411 | IF(1:100) |
| Peptide, recombinant protein | AlexaFluor 488 Phalloidin | Life Technologies | Cat#:A12379 | IF cells 3D (1:50) |
| Peptide, recombinant protein | Streptavidin-488 | Invitrogen | Cat#:S11223 | 25 µg/ml |
| Peptide, recombinant protein | 10 kDa Dextran-Texas Red | Invitrogen | Cat#:D1863 | 100 µl injected at 5 mg/ml |
| Peptide, recombinant protein | Fibrinogen | Fisher | Cat#:820224 | 500 µl at 2.2 mg/ml |
| Peptide, recombinant protein | Fibronectin | Sigma | Cat#:F2006-2MG | 5 µg/ml |
| Peptide, recombinant protein | EZ-Link Sulfo-NHS-LC-Biotin | Thermo Fisher | Cat#:A39257 | 0.5 mM |
| Chemical compound, drug | Tamoxifen | Sigma | Cat#:T5648 | 50 µl injected at 1 mg/ml |
| Chemical compound, drug | Thrombin | Sigma | Cat#:T7201-500UN | For bead assay: 7 µl at 50 U/ml For cell treatments: 0.5 U/ml for 10 min at 37°C |
| Chemical compound, drug | (-) Blebbistatin | Sigma | B0560-1MG | 10 µM for 15 min at 37°C |
| Chemical compound, drug | Y-27632 | VWR | Cat#:10187-694 | 10 µM for 30 min at 37°C |
| Chemical compound, drug | Nocodazole | Sigma | Cat#:M1404 | 10 µM for 20 min at 37°C |

*Appendix 1 Continued on next page*

*Appendix 1 Continued*

| Reagent type (species) or resource | Designation | Source or reference | Identifiers | Additional information |
|---|---|---|---|---|
| Chemical compound, drug | EDTA | Sigma-Aldrich | Cat#:EDS-100G | 3 mM |
| Commercial assay or kit | KAPA mRNA HyperPrep Kit | Roche | Cat#:7961901001 | |
| Commercial assay or kit | Click-It EdU Kit 488 | Invitrogen | Cat#:C10337 | |
| Software, algorithm | Fiji | *Linkert et al., 2010* (PMID: 20513764); *Schindelin et al., 2012* (PMID: 22743772) | | |
| Software, algorithm | STAR | *Dobin et al., 2013* (PMID: 23104886) | | |
| Software, algorithm | HTSeq 2.0 | *Putri et al., 2022* (PMID: 35561197) | | |
| Software, algorithm | DESeq2 | *Love et al., 2014* (PMID: 25516281) | | |
| Software, algorithm | Visual Basic algorithm for tip tracking | Other | | Dr Dan Buster |
| Software, algorithm | Prism 9 | GraphPad | | |
| Software, algorithm | Fluoview FV31S-SW | Olympus | | |
| Software, algorithm | MetaMorph | MetaMorph | | |
| Other | DAPI | Sigma | Cat#:10236276001 | DNA stain, 0.3 µM |
| Other | DRAQ7 | Abcam | Cat#:ab109202 | DNA stain, 1:1000 |
| Other | xCELLigence Real-Time Cell Analyzer | Acea Biosciences/ Roche Applied Science | | Equipment to assess electrical resistance across cell monolayer |
| Other | Ibidi pump system | Ibidi | Cat#:10902 | Pump system to generate laminar flow across cells |

