## [Editor Report]

Endothelial cells lining the inner side of blood vessels constitute the blood-tissue barrier via regulation of cell-cell junctions. The cell nucleus regulates endothelial cell behaviors, but it is unclear how the nucleus contributes to endothelial cell activities at the cell membrane. This study for the first time demonstrates that nuclear membrane protein SUN1 stabilizes endothelial cell-cell junctions far from the nucleus via regulation of microtubule dynamics and Rho GEF-H1 signaling, revealing long-range cellular communication important for vascular development and endothelial barrier function.

---

## [Decision Letter]

**Decision letter after peer review:**

Thank you for submitting your article "Nuclear SUN1 stabilizes endothelial cell junctions via microtubules to regulate blood vessel formation" for consideration by *eLife*. Your article has been reviewed by 3 peer reviewers, and the evaluation has been overseen by a Reviewing Editor and Anna Akhmanova as the Senior Editor. The reviewers have opted to remain anonymous.

Essential revisions:

The reviewers agreed that the data on the role of Sun1 in junctional organization and angiogenic sprouting are novel and interesting, but also thought that there were some substantial weaknesses, particularly due to the lack of specific controls and the need for additional support for some of the conclusions. The reviewers also thought that the presentation can be substantially improved and made various detailed suggestions on writing, which can be found in their Recommendations for the authors. We advise you to consider these points seriously.

I believe the authors can revise the manuscript successfully given their expertise, but please provide the reasons for not implementing the suggested changes if necessary.

*Reviewer #1 (Recommendations for the authors):*

Buglak et al. describe a role for the nuclear envelope protein Sun1 in endothelial mechanotransduction and vascular development. The study provides a full mechanistic investigation of how Sun1 is achieving its function, which supports the concept that nuclear anchoring is important for proper mechanosensing and junctional organization. The experiments have been well designed and were quantified based on independent experiments. The experiments are convincing and of high quality and include Sun1 depletion in endothelial cell cultures, zebrafish, and in endothelial-specific inducible knockouts in mice.

There are a few comments that the authors need to address:

– Figure 1E. The endothelial-specific depletion of Sun1 is affecting the endothelial junctions. In the provided image it seems as if the total intensity of VE-cadherin staining is increased. Is this a result that is recapitulated in other samples too?

– Figure 3I shows that there is an abnormal junction in the developing ISVs of Sun1 MO fish. Given the notion that the ISVs develop differently, this is perhaps not so unexpected. Perhaps the ISVS of the WT fish are already perfused and the MO not yet since they are lacking behind. Are there any differences in junctional interactions in ISVS that are comparable in length?

– Regarding the VE-cadherin internalization assay. There is often a large variety of internalization in endothelial cultures. Can the authors provide a larger field of view for readers to indicate in what proportion of cells the increasing internalization levels are observed?

– It remains difficult to conclude whether the observed changes upon Sun1 depletion on vascular development are dependent on its effect on junctions, or whether they may involve other microtubule-associating cellular structures as well. For instance the focal adhesions, or organization of the actomyosin cytoskeleton itself. Is there another reason perhaps that strengthens the conclusion that Sun1 mediates its effect mostly via the junctions? Or do the authors think that other cellular structures contribute as well?

*Reviewer #2 (Recommendations for the authors):*

Endothelial cells mediate the growth of the vascular system but they also need to prevent vascular leakage, which involves interactions with neighboring endothelial cells (ECs) through junctional protein complexes. Buglak et al. report that the EC nucleus controls the function of cell-cell junctions through the nuclear envelope-associated proteins SUN1 and Nesprin-1. They argue that SUN1 controls microtubule dynamics and junctional stability through the RhoA activator GEF-H1.

In my view, this study is interesting and addresses an important but very little-studied question, namely the link between the EC nucleus and cell junctions in the periphery. The study has also made use of different model systems, i.e. genetically modified mice, zebrafish, and cultured endothelial cells, which confirms certain findings and utilizes the specific advantages of each model system. A weakness is that some important controls are missing. In addition, the evidence for the proposed molecular mechanism should be strengthened.

Specific comments:

1) Data showing the efficiency of Sun1 inactivation in the murine endothelial cells is lacking. It would be best to see what is happening on the protein level, but it would already help a great deal if the authors could show a reduction of the transcript in sorted ECs. The excision of a DNA fragment shown in the lung (Figure 1—figure supplement 1C) is not quantitative at all. In addition, the gel has been run way too short so it is impossible to even estimate the size of the DNA fragment.

2) The authors show an increase in vessel density in the periphery of the growing Sun1 mutant retinal vasculature. It would be important to add staining with a marker labelling EC nuclei (e.g. Erg) because higher vessel density might reflect changes in cell size/shape or number, which has also implications for the appearance of cell-cell junctions. More ECs crowded within a small area are likely to have more complicated junctions.

Furthermore, it would be useful and straightforward to assess EC proliferation, which is mentioned later in the experiments with cultured ECs but has not been addressed in the in vivo part.

3) It appears that the loss of Sun1/sun1b in mice and zebrafish is compatible with major aspects of vascular growth and leads to changes in filopodia dynamics and vascular permeability (during development) without severe and lasting disruption of the EC network. It would be helpful to know whether the loss-of-function mutants can ultimately form a normal vascular network in the retina and trunk, respectively. It might be sufficient to mention this in the text.

4) The only readout after the rescue of the SUN1 knockdown by GEF-H1 depletion is the appearance of VE-cadherin+ junctions (Figure 6G and H). This is insufficient evidence for a relatively strong conclusion. The authors should at least look at microtubules. They might also want to consider the activation status of RhoA as a good biochemical readout. It is argued that RhoA activity goes up (see Figure 7C) but there is no data supporting this conclusion. It is also not clear whether "diffuse" GEF-H1 localization translates into increased Rho A activity, as is suggested by the Rho kinase inhibition experiment. GEF-H1 levels in the Western blot in (Figure 6- supplement 2C) have not been quantitated.

5) The criticism raised for the GEF-H1 rescue also applies to the co-depletion of SUN1 and Nesprin-1. This mechanistic aspect is currently somewhat weak and should be strengthened. Again, Rho A activity might be a useful and quantitative biochemical readout.

6) Likewise, I have seen no data measuring VE-cadherin internalization and therefore the statement in line 309 should be modified.

*Reviewer #3 (Recommendations for the authors):*

Here, Buglak and coauthors describe the effect of Sun1 deficiency on endothelial junctions. Sun1 is a component of the LINC complex, connecting the inner nuclear membrane with the cytoskeleton. The authors show that in the absence of Sun1, the morphology of the endothelial adherens junction protein VE-cadherin is altered, indicative of increased internalization of VE-cadherin. The change in VE-cadherin dynamics correlates with decreased angiogenic sprouting as shown using in vivo and in vitro models. The study would benefit from a stricter presentation of the data and needs additional controls in certain analyses.

1. The authors implicate the changes in VE-cadherin morphology to be of consequence for "barrier function" and mention barrier function frequently throughout the text, for example in the heading on page 12: "SUN1 stabilizes endothelial cell-cell junctions and regulates barrier function". The concept of "barrier" implies the ability of endothelial cells to restrict the passage of molecules and cells across the vessel wall. This is tested only marginally (Suppl Figure 1F) and these data are not quantified. Increased leakage of 10kDa dextran in a P6-7 Sun1-deficient retina as shown here probably reflects the increased immaturity of the Sun1-deficient retinal vasculature. From these data, the authors cannot state that Sun1 regulates the barrier or barrier function (unclear what exactly the authors refer to when they make a distinction between the barrier as such on the one hand and barrier function on the other). The authors can, if they do more experiments, state that loss of Sun1 leads to increased leakage in the early postnatal stages in the retina. However, if they wish to characterize the vascular barrier, there is a wide range of other tissue that should be tested, in the presence and absence of disease. Moreover, a regulatory role for Sun1 would imply that Sun1 normally, possibly through changes in its expression levels, would modulate the barrier properties to allow more or less leakage in different circumstances. However, no such data are shown. The authors would need to go through their paper and remove statements regarding the regulation of the barrier and barrier function since these are conclusions that lack foundation.

2. In Figure 6g, the authors show that "depletion of GEF-H1 in endothelial cells that were also depleted for SUN1 rescued the destabilized cell-cell junctions observed with SUN1 KD alone". However, it is quite clear that Sun1 depletion also affects cell shape and cell alignment and this is not rescued by GEF-H1 depletion (Figure 6g). This should be described and commented on. Moreover please show the effects of GEF-H1 alone.

3. In Figure 6a, the authors show rescue of junction morphology in Sun1-depleted cells by deletion of Nesprin1. The effect of Nesprin1 KD alone is missing.

Buglak et al., describe the effect of deleting Sun1 from endothelial cells in vivo and in vitro. They observe changes in VE-cadherin morphology and note effects on VE-cadherin internalization, which they suggest explains the reduced angiogenic sprouting capacity in the absence of Sun1. The authors present evidence that reducing GEF-H1 or Nesprin1, can partially rescue the loss of the Sun1 phenotype. There is a precedence of data in the literature showing regulation of cytoskeleton (both microtubule and actin) by the LINC complex, in which both Sun1 and Nesprin1 are part, in a range of cell types. However, the consequence of Sun1 deficiency on endothelial adherens junctions has not been described before. Unfortunately, the authors present their data in a rather indistinctive manner, using inaccurate expressions and there is also a tendency for overinterpretation of the data. Moreover, by focusing on endothelial cells and their unique barrier to blood components, specificity is implied which is not true in well-known biology.

4. The penetration of streptavidin between cells in the siRNA-treated HUVECs and the impedance changes with Sun1-deletion is in agreement with Sun1 being a structural component in a mechanosensitive microtubule- and actin-dependent cell-matrix or cell-cell adhesion. See Mol Biol Cell. 2021 Aug 19;32(18):1654-1663. doi: 10.1091/mbc.E20-11-0698. This paper should be cited. Please also check the paper by Ueda et al., in Front Cell Dev Biol. 2022 May 18;10:885859. doi: 10.3389/fcell.2022.885859, which describes Sun1 as part of the mechanosensing machinery, which is most likely the underlying mechanism explaining the effects recorded by Buglak et al. in the various analyses. This paper by Ueda is in the reference list but strangely not in the text as far as I have been able to see.

5. The authors have chosen the morphology of VE-cadherin as a readout for the effect of Sun1-deficiency on endothelial cells. They describe the effect of Sun1 depletion as "junction activation" and even describe "over-activated" junctions (line 241). This wording implies the involvement of enzymatic activity and the impression may be that the authors suggest that VE-cadherin is an enzyme. This is not correct. Junctions can be more or less stable as a consequence of the interruption of VE-cadherin homophilic bonds and internalization, for which the authors present data. The authors would do better to use less ambiguous language.

6. Throughout the paper please change the "control" designation for the non-floxed mouse strain to the actual genotype. As the authors are aware, Cre-toxicity can cause very similar effects on the retina vasculature as shown for the Sun1 iECKO. This is also apparent from Supplemental Figure 1. Therefore, it is important to indicate that Cre+ wildtype mice have been used as controls rather than a Cre- wildtype.

7. What is the effect of Sun1 depletion on endothelial migration and proliferation?

---

## [Author Response]

Essential revisions:The reviewers agreed that the data on the role of Sun1 in junctional organization and angiogenic sprouting are novel and interesting, but also thought that there were some substantial weaknesses, particularly due to the lack of specific controls and the need for additional support for some of the conclusions. The reviewers also thought that the presentation can be substantially improved and made various detailed suggestions on writing, which can be found in their Recommendations for the authors. We advise you to consider these points seriously.I believe the authors can revise the manuscript successfully given their expertise, but please provide the reasons for not implementing the suggested changes if necessary.

We thank the editors and reviewers for their enthusiasm for our work and for their insightful feedback. The reviewers agreed that our work investigating the LINC complex protein Sun1 in junction organization and function and angiogenic sprouting is novel and interesting, and they also asked for revisions, including some additional controls and more support for some conclusions. The editor asked us to specifically focus on points raised in the public review of Reviewer 2 (particularly points 4 and 5, as well as requests for controls raised in points 1 and 2) and the three points raised in the public review of Reviewer 3. The reviewers also made various detailed suggestions on the writing.

We have seriously considered these points and present a revised manuscript with new data that addresses reviewer concerns and substantially strengthens the conclusions. Specific new data (new panels: Figure 1-figure supplement 1F-G; Figure 6I-J; Figure 6-figure supplement 3B-C; Figure 7B-C, E-F; Figure 7-figure supplement 1B-G) now extends our analysis of the rescue of SUN1-depleted endothelial junctions upon co-depletion of GEFH1 or NESP1 to microtubule parameters and a functional readout of junction integrity. We also provide direct evidence of RhoA over-activation with SUN1 depletion and rescue by GEFH1 co-depletion via Rho activity pulldowns (Figure 6-figure supplement 2D). We’ve also modified several panels in response to reviewer comments (Figure 1B, 1E; Figure 3J; Figure 4-figure supplement 1B, 1F; Figure 6-figure supplement 3A). Finally, we’ve made text changes to clarify our results and put our work into context.

Reviewer #1 (Recommendations for the authors):Buglak et al. describe a role for the nuclear envelope protein Sun1 in endothelial mechanotransduction and vascular development. The study provides a full mechanistic investigation of how Sun1 is achieving its function, which supports the concept that nuclear anchoring is important for proper mechanosensing and junctional organization. The experiments have been well designed and were quantified based on independent experiments. The experiments are convincing and of high quality and include Sun1 depletion in endothelial cell cultures, zebrafish, and in endothelial-specific inducible knockouts in mice.

We thank the reviewer for their enthusiastic comments and for noting our use of multiple model systems.

There are a few comments that the authors need to address:– Figure 1E. The endothelial-specific depletion of Sun1 is affecting the endothelial junctions. In the provided image it seems as if the total intensity of VE-cadherin staining is increased. Is this a result that is recapitulated in other samples too?

We thank the reviewer for pointing this out. Upon re-examination of the data, we do not believe there is a change in VE-cadherin intensity. We have updated the figure panel in Figure 1E to a more representative image. We have also indicated the specific genotypes used in the figure panels.

– Figure 3I shows that there is an abnormal junction in the developing ISVs of Sun1 MO fish. Given the notion that the ISVs develop differently, this is perhaps not so unexpected. Perhaps the ISVS of the WT fish are already perfused and the MO not yet since they are lacking behind. Are there any differences in junctional interactions in ISVS that are comparable in length?

The reviewer makes a good point. Given that we see aberrant connections of ISVs at the DLAV in Sun1 MO fish, we have gone back and reanalyzed the junctions binned by the ISV that reach the DLAV vs. those that do not connect to DLAV (Figure 3J). We find that the ISV junctions in Sun1 MO fish are significantly less linear regardless of DLAV anastomosis.

– Regarding the VE-cadherin internalization assay. There is often a large variety of internalization in endothelial cultures. Can the authors provide a larger field of view for readers to indicate in what proportion of cells the increasing internalization levels are observed?

We agree with the reviewer that VE-cadherin internalization can be somewhat heterogenous within a culture. We now include a larger field of view in Figure 4—figure supplement 1F to show multiple cells.

– It remains difficult to conclude whether the observed changes upon Sun1 depletion on vascular development are dependent on its effect on junctions, or whether they may involve other microtubule-associating cellular structures as well. For instance the focal adhesions, or organization of the actomyosin cytoskeleton itself. Is there another reason perhaps that strengthens the conclusion that Sun1 mediates its effect mostly via the junctions? Or do the authors think that other cellular structures contribute as well?

We agree with the reviewer that other cellular processes may contribute to SUN1 effects on vascular development, although our work shows a clear effect of SUN1 depletion on junctions and links to microtubules through dynamics and junction contacts, and these conclusions are supported by new data. A recent study found that SUN1 regulates focal adhesions via the actin cytoskeleton in non-endothelial cells (Ueda et al., 2022) and other studies show LINC complex regulation of endothelial migration (Denis et al., 2021; King et al., 2014). Both cell-matrix adhesions and cell migration are important for vascular development, and we now bring these ideas into the Discussion (p. 21).

Reviewer #2 (Recommendations for the authors):Endothelial cells mediate the growth of the vascular system but they also need to prevent vascular leakage, which involves interactions with neighboring endothelial cells (ECs) through junctional protein complexes. Buglak et al. report that the EC nucleus controls the function of cell-cell junctions through the nuclear envelope-associated proteins SUN1 and Nesprin-1. They argue that SUN1 controls microtubule dynamics and junctional stability through the RhoA activator GEF-H1.In my view, this study is interesting and addresses an important but very little-studied question, namely the link between the EC nucleus and cell junctions in the periphery. The study has also made use of different model systems, i.e. genetically modified mice, zebrafish, and cultured endothelial cells, which confirms certain findings and utilizes the specific advantages of each model system. A weakness is that some important controls are missing. In addition, the evidence for the proposed molecular mechanism should be strengthened.

We thank the reviewer for their interest in our work and for highlighting the relative lack of information regarding connections between the EC nucleus and cell periphery, and for noting our use of multiple model systems. We thank the reviewer for suggesting additional controls and mechanistic support, and we have made the revisions described below.

Specific comments:1) Data showing the efficiency of Sun1 inactivation in the murine endothelial cells is lacking. It would be best to see what is happening on the protein level, but it would already help a great deal if the authors could show a reduction of the transcript in sorted ECs. The excision of a DNA fragment shown in the lung (Figure 1—figure supplement 1C) is not quantitative at all. In addition, the gel has been run way too short so it is impossible to even estimate the size of the DNA fragment.

We agree that the DNA excision is not sufficient to demonstrate excision efficiency. We attempted examination of SUN1 protein levels in mutant retinas via immunofluorescence, but to date we have not found a SUN1 antibody that works in mouse retinal explants. We argue that mouse EC isolation protocols enrich but don’t give 100% purity, so that RNA analysis of lung tissue also has caveats. Finally, we contend that our demonstration of a consistent vascular phenotype in *Sun1^iECKO^* mutant retinas argues that excision has occurred. To test the efficiency of our excision protocol, we bred *Cdh5CreERT2* mice with the *ROSA^mT/mG^* excision reporter (cells express tdTomato absent Cre activity and express GFP upon Cre-mediated excision; Muzumdar et al., 2007). Utilizing the same excision protocol as used for the *Sun1^iECKO^* mice, we see a significantly high level of excision in retinal vessels only in the presence of *Cdh5CreERT2* (Author response image 1).

**Author response image 1. sa2fig1:** *Cdh5CreERT2* efficiently excises in endothelial cells of the mouse postnatal retina. (**A**) Representative images of P7 mouse retinas with the indicated genotypes, stained for ERG (white, nucleus). tdTomato (magenta) is expressed in cells that have not undergone Cre-mediated excision, while GFP (green) is expressed in excised cells. Scale bar, 100μm. (**B**) Quantification of tdTomato fluorescence relative to GFP fluorescence as shown in A. tdTomato and GFP fluorescence of endothelial cells was measured by creating a mask of the ERG channel. n=3 mice per genotype. ***, *p*<0.001 by student’s two-tailed unpaired *t*-test.

2) The authors show an increase in vessel density in the periphery of the growing Sun1 mutant retinal vasculature. It would be important to add staining with a marker labelling EC nuclei (e.g. Erg) because higher vessel density might reflect changes in cell size/shape or number, which has also implications for the appearance of cell-cell junctions. More ECs crowded within a small area are likely to have more complicated junctions.Furthermore, it would be useful and straightforward to assess EC proliferation, which is mentioned later in the experiments with cultured ECs but has not been addressed in the in vivo part.

We concur that ERG staining is important to show any changes in nuclear shape or cell density in the post-natal retina. We now include this data in Figure1—figure supplement 1F-G. We do not see obvious changes in nuclear shape or number, though we do observe some crowding in *Sun1^iECKO^* retinas, consistent with increased density. However, when normalized to total vessel area, we do not observe a significant difference in the nuclear signal density in *Sun1^iECKO^* mutant retinas relative to controls.

3) It appears that the loss of Sun1/sun1b in mice and zebrafish is compatible with major aspects of vascular growth and leads to changes in filopodia dynamics and vascular permeability (during development) without severe and lasting disruption of the EC network. It would be helpful to know whether the loss-of-function mutants can ultimately form a normal vascular network in the retina and trunk, respectively. It might be sufficient to mention this in the text.

We thank the reviewer for pointing this out. It is true that developmental defects in the vasculature resulting from various genetic mutations are often resolved over time.

We’ve made text changes to discuss viability of Sun1 global KO mice and lack of perduring effects in sun1 morphant fish, perhaps resulting from compensation by SUN2, which is partially functionally redundant with SUN1 in vivo (Lei et al., 2009; Zhang, et al., 2009) (p. 20).

4) The only readout after the rescue of the SUN1 knockdown by GEF-H1 depletion is the appearance of VE-cadherin+ junctions (Figure 6G and H). This is insufficient evidence for a relatively strong conclusion. The authors should at least look at microtubules. They might also want to consider the activation status of RhoA as a good biochemical readout. It is argued that RhoA activity goes up (see Figure 7C) but there is no data supporting this conclusion. It is also not clear whether "diffuse" GEF-H1 localization translates into increased Rho A activity, as is suggested by the Rho kinase inhibition experiment. GEF-H1 levels in the Western blot in (Figure 6- supplement 2C) have not been quantitated.

We agree that analysis of RhoA activity and additional analysis of rescued junctions strengthens our conclusions, so we performed these experiments. New data (Figure 6IJ) shows that co-depletion of SUN1 and GEF-H1 rescues junction integrity as measured by biotin-matrix labeling. Interestingly, co-depletion of SUN1 and GEF-H1 does not rescue reduced microtubule density at the periphery (Figure 6—figure supplement 3BC), placing GEF-H1 downstream of aberrant microtubule dynamics in SUN1 depleted cells. This is consistent with our model (Figure 8) describing how loss of SUN1 leads to increased microtubule depolymerization, resulting in release and activation of GEF-H1 that goes on to affect actomyosin contractility and junction integrity. In addition, we include images of the junctions in GEF-H1 single KD (Figure 6—figure supplement 3BC) and quantify the western blot in Figure 6—figure supplement 3A.

We performed RhoA activity assays and new data shows that SUN1 depletion results in increased RhoA activation, while co-depletion of SUN1 and GEF-H1 ameliorates this increase (Figure 6—figure supplement 2D). This is consistent with our model in which loss of SUN1 leads to increased RhoA activity via release of GEF-H1 from microtubules. In addition, we now cite a recent study describing that GEF-H1 is activated when unbound to microtubules, with this activation resulting in increased RhoA activity (Azoitei et al., 2019).

5) The criticism raised for the GEF-H1 rescue also applies to the co-depletion of SUN1 and Nesprin-1. This mechanistic aspect is currently somewhat weak and should be strengthened. Again, Rho A activity might be a useful and quantitative biochemical readout.

We respectfully point out that we showed that co-depletion of nesprin-1 and SUN1 rescues SUN1 knockdown effects via several readouts, including rescue of junction morphology, biotin labeling, microtubule localization at the periphery, and GEFH1/microtubule localization. We’ve moved this data to the main figure (Figure 7B-C, E-F) to better highlight these mechanistic findings. These results are consistent with our model that nesprin-1 effects are upstream of GEF-H1 localization. We also added results showing that nesprin-1 knockdown alone does not affect junction integrity, microtubule density, or GEF-H1/microtubule localization (Figure 7—figure supplement 1B-G).

6) Likewise, I have seen no data measuring VE-cadherin internalization and therefore the statement in line 309 should be modified.

We respectfully direct the reviewer to Figure 4—figure supplement 1F-G showing increased VE-cadherin internalization in SUN1-depleted endothelial cells. We updated this panel to include a larger field of view to show a more representative cell sampling.

Reviewer #3 (Recommendations for the authors):Here, Buglak and coauthors describe the effect of Sun1 deficiency on endothelial junctions. Sun1 is a component of the LINC complex, connecting the inner nuclear membrane with the cytoskeleton. The authors show that in the absence of Sun1, the morphology of the endothelial adherens junction protein VE-cadherin is altered, indicative of increased internalization of VE-cadherin. The change in VE-cadherin dynamics correlates with decreased angiogenic sprouting as shown using in vivo and in vitro models. The study would benefit from a stricter presentation of the data and needs additional controls in certain analyses.

We thank the reviewer for their insightful comments, and in response we have performed the revisions described below.

1. The authors implicate the changes in VE-cadherin morphology to be of consequence for "barrier function" and mention barrier function frequently throughout the text, for example in the heading on page 12: "SUN1 stabilizes endothelial cell-cell junctions and regulates barrier function". The concept of "barrier" implies the ability of endothelial cells to restrict the passage of molecules and cells across the vessel wall. This is tested only marginally (Suppl Figure 1F) and these data are not quantified. Increased leakage of 10kDa dextran in a P6-7 Sun1-deficient retina as shown here probably reflects the increased immaturity of the Sun1-deficient retinal vasculature. From these data, the authors cannot state that Sun1 regulates the barrier or barrier function (unclear what exactly the authors refer to when they make a distinction between the barrier as such on the one hand and barrier function on the other). The authors can, if they do more experiments, state that loss of Sun1 leads to increased leakage in the early postnatal stages in the retina. However, if they wish to characterize the vascular barrier, there is a wide range of other tissue that should be tested, in the presence and absence of disease. Moreover, a regulatory role for Sun1 would imply that Sun1 normally, possibly through changes in its expression levels, would modulate the barrier properties to allow more or less leakage in different circumstances. However, no such data are shown. The authors would need to go through their paper and remove statements regarding the regulation of the barrier and barrier function since these are conclusions that lack foundation.

We thank the reviewer for pointing out that the language used regarding the function and integrity of the junctions is confusing, although we suggest that the endothelial cell properties measured by our assays are typically equated with “barrier function” in the literature. However, we have edited our language to precisely describe our results as suggested by the reviewer.

2. In Figure 6g, the authors show that "depletion of GEF-H1 in endothelial cells that were also depleted for SUN1 rescued the destabilized cell-cell junctions observed with SUN1 KD alone". However, it is quite clear that Sun1 depletion also affects cell shape and cell alignment and this is not rescued by GEF-H1 depletion (Figure 6g). This should be described and commented on. Moreover please show the effects of GEF-H1 alone.

We thank the reviewer for pointing out the effects on cell shape. SUN1 depletion typically leads to shape changes consistent with elevated contractility, but this is considered to be downstream of the effects quantified here. We updated the panel in Figure 6G to a more representative image showing cell shape rescue by co-depletion of SUN1 and GEF-H1. We present new data panels showing that GEF-H1 depletion alone does not affect junction integrity (Figure 6I-J). We also present new data showing that co-depletion of GEF-H1 and SUN1 does not rescue microtubule density at the periphery (Figure 6—figure supplement 3B-C), consistent with our model that GEF-H1 activation is downstream of microtubule perturbations induced by SUN1 loss.

3. In Figure 6a, the authors show rescue of junction morphology in Sun1-depleted cells by deletion of Nesprin1. The effect of Nesprin1 KD alone is missing.

We thank the reviewer for this comment, and we now include new panels (Figure 7figure supplement 1B-G) demonstrating that Nesprin-1 depletion does not affect biotin-matrix labeling, peripheral microtubule density, or GEF-H1/microtubule localization absent co-depletion with SUN1. These findings are consistent with our model that Nesprin-1 loss does not affect cell junctions on its own because it is held in a non-functional complex with SUN1 that is not available in the absence of SUN1.

Buglak et al., describe the effect of deleting Sun1 from endothelial cells in vivo and in vitro. They observe changes in VE-cadherin morphology and note effects on VE-cadherin internalization, which they suggest explains the reduced angiogenic sprouting capacity in the absence of Sun1. The authors present evidence that reducing GEF-H1 or Nesprin1, can partially rescue the loss of the Sun1 phenotype. There is a precedence of data in the literature showing regulation of cytoskeleton (both microtubule and actin) by the LINC complex, in which both Sun1 and Nesprin1 are part, in a range of cell types. However, the consequence of Sun1 deficiency on endothelial adherens junctions has not been described before. Unfortunately, the authors present their data in a rather indistinctive manner, using inaccurate expressions and there is also a tendency for overinterpretation of the data. Moreover, by focusing on endothelial cells and their unique barrier to blood components, specificity is implied which is not true in well-known biology.

We thank the reviewer for their insight. While we agree that cytoskeleton regulation by the LINC complex has been studied, we contend that most published studies disrupt the LINC complex as a whole (via dominant negative constructs or double knockdowns to disrupt multiple LINC components). Thus, relatively little is known regarding functions of individual LINC components, or how different combinations of SUN and KASH proteins may beget different cellular functions. Thus, our goal was to examine the specific role of SUN1 in the regulation of cell junctions and cytoskeleton in the vasculature. We agree that the interactions and mechanisms that we uncovered are likely not specific to endothelial cells, and we now discuss that our findings may be applicable to other cell and tissue types (p. 19-20).

4. The penetration of streptavidin between cells in the siRNA-treated HUVECs and the impedance changes with Sun1-deletion is in agreement with Sun1 being a structural component in a mechanosensitive microtubule- and actin-dependent cell-matrix or cell-cell adhesion. See Mol Biol Cell. 2021 Aug 19;32(18):1654-1663. doi: 10.1091/mbc.E20-11-0698. This paper should be cited. Please also check the paper by Ueda et al., in Front Cell Dev Biol. 2022 May 18;10:885859. doi: 10.3389/fcell.2022.885859, which describes Sun1 as part of the mechanosensing machinery, which is most likely the underlying mechanism explaining the effects recorded by Buglak et al. in the various analyses. This paper by Ueda is in the reference list but strangely not in the text as far as I have been able to see.

We respectfully submit that we did discuss the paper by Ueda et al. and the paper by Denis et al. (p. 6, 9, 21). We’ve now expanded this discussion to better highlight those studies, and we now comment that additional cellular processes may be affected by SUN1 (p. 13, 21).

5. The authors have chosen the morphology of VE-cadherin as a readout for the effect of Sun1-deficiency on endothelial cells. They describe the effect of Sun1 depletion as "junction activation" and even describe "over-activated" junctions (line 241). This wording implies the involvement of enzymatic activity and the impression may be that the authors suggest that VE-cadherin is an enzyme. This is not correct. Junctions can be more or less stable as a consequence of the interruption of VE-cadherin homophilic bonds and internalization, for which the authors present data. The authors would do better to use less ambiguous language.

We agree with the reviewer that the wording was confusing, and we have updated the text to describe “destabilized” rather than “activated” junctions.

6. Throughout the paper please change the "control" designation for the non-floxed mouse strain to the actual genotype. As the authors are aware, Cre-toxicity can cause very similar effects on the retina vasculature as shown for the Sun1 iECKO. This is also apparent from Supplemental Figure 1. Therefore, it is important to indicate that Cre+ wildtype mice have been used as controls rather than a Cre- wildtype.

Our “control” designation represents a combination of genotypes from littermate controls as described in Materials and methods and includes WT, Cdh5CreERT2/+, Sun1^fl/+^, and Sun1^fl/fl^ mice. Genotypes were combined for quantification of controls due to small litter sizes. We agree that Cre-toxicity can cause similar effects on the retina. It is for this reason that we separated out retinas by genotype in Figure 1—figure supplement 1D. We include that graph in Author response image 2 with additional statistical comparisons showing that there is no significant difference between Cdh5CreERT2/+ retinas and WT retinas, but a significant difference between Cdh5CreERT2/+ retinas and Sun1^iECKO^ retinas. We agree with the reviewer that it is important to show the phenotype for the Cre+ mice, so we have updated Figure 1B and Figure 1E to pair Cdh5CreERT2/+ retinas with Sun1^iECKO^ retinas. We find that Cdh5CreERT2/+ retinas have normal radial expansion and density and do not display disorganized VE-cadherin staining like Sun1^iECKO^ retinas.

**Author response image 2. sa2fig2:** Cdh5CreERT2 does not affect radial expansion. Reproduced graph from Figure 1—figure supplement 1D with additional statistical comparisons. There is no significant difference in radial expansion between Cdh5CreERT2/+ mice and WT mice.

7. What is the effect of Sun1 depletion on endothelial migration and proliferation?

We show that there is no significant difference in proliferation following SUN1 depletion in endothelial monolayers by Ki67 staining and EdU incorporation. Our focus was on endothelial cell junction function and not migration; however, since LINC complex manipulations via KASH dominant negative or nesprin-1 or nesprin-2 KD lead to reduced endothelial cell migration (Denis et al., 2021; King et al., 2014), and we document a sprout retraction phenotype, we now discuss possible effects of SUN1 loss on migration in the text (p. 21).

References

Azoitei, M. L., Noh, J., Marston, D. J., Roudot, P., Marshall, C. B., Daugird, T. A., Lisanza, S. L., Sandί, M., Ikura, M., Sondek, J., Rottapel, R., Hahn, K. M., Danuser, and Danuser, G. (2019). Spatiotemporal dynamics of GEF-H1 activation controlled by microtubule- and Src-mediated pathways. Journal of Cell Biology, 218(9), 3077-3097. https://doi.org/10.1083/jcb.201812073

Denis, K. B., Cabe, J. I., Danielsson, B. E., Tieu, K. V, Mayer, C. R., and Conway, D. E. (2021). The LINC complex is required for endothelial cell adhesion and adaptation to shear stress and cyclic stretch. Molecular Biology of the Cell, mbcE20110698. https://doi.org/10.1091/mbc.E20-11-0698

King, S. J., Nowak, K., Suryavanshi, N., Holt, I., Shanahan, C. M., and Ridley, A. J. (2014). Nesprin-1 and nesprin-2 regulate endothelial cell shape and migration. Cytoskeleton (Hoboken, N.J.), 71(7), 423–434. https://doi.org/10.1002/cm.21182

Lei, K., Zhang, X., Ding, X., Guo, X., Chen, M., Zhu, B., Xu, T., Zhuang, Y., Xu, R., and Han, M. (2009). SUN1 and SUN2 play critical but partially redundant roles in anchoring nuclei in skeletal muscle cells in mice. PNAS, 106(25), 10207–10212.

Muzumdar, M. D., Tasic, B., Miyamichi, K., Li, L., and Luo, L. (2007). A global double fluorescent Cre reporter mouse. Genesis, 45(9), 593-605. https://doi.org/10.1002/dvg.20335

Ueda, N., Maekawa, M., Matsui, T. S., Deguchi, S., Takata, T., Katahira, J., Higashiyama, S., and Hieda, M. (2022). Inner Nuclear Membrane Protein, SUN1, is Required for Cytoskeletal Force Generation and Focal Adhesion Maturation. Frontiers in Cell and Developmental Biology, 10, 885859. https://doi.org/10.3389/fcell.2022.885859

Zhang, X., Lei, K., Yuan, X., Wu, X., Zhuang, Y., Xu, T., Xu, R., and Han, M. (2009). SUN1/2 and Syne/Nesprin-1/2 complexes connect centrosome to the nucleus during neurogenesis and neuronal migration in mice. Neuron, 64(2), 173–187. https://doi.org/10.1016/j.neuron.2009.08.018.